# In situ functional dissection of RNA *cis*-regulatory elements by multiplex CRISPR-Cas9 genome engineering

Qianxin Wu[1,11], Quentin R.V. Ferry[1], Toni A. Baeumler[1], Yale S. Michaels[1], Dimitrios M. Vitsios [2], Omer Habib[3], Roland Arnold[4], Xiaowei Jiang [4], Stefano Maio[5], Bruno R. Steinkraus[1], Marta Tapia[6], Paolo Piazza[7], Ni Xu[1], Georg A. Holländer[5,8], Thomas A. Milne [6], Jin-Soo Kim[3,9], Anton J. Enright [2], Andrew R. Bassett [10,11] & Tudor A. Fulga[1]

RNA regulatory elements (RREs) are an important yet relatively under-explored facet of gene regulation. Deciphering the prevalence and functional impact of this post-transcriptional control layer requires technologies for disrupting RREs without perturbing cellular homeostasis. Here we describe genome-engineering based evaluation of RNA regulatory element activity (GenERA), a clustered regularly interspaced short palindromic repeats (CRISPR)-Cas9 platform for in situ high-content functional analysis of RREs. We use GenERA to survey the entire regulatory landscape of a 3'UTR, and apply it in a multiplex fashion to analyse combinatorial interactions between sets of miRNA response elements (MREs), providing strong evidence for cooperative activity. We also employ this technology to probe the functionality of an entire MRE network under cellular homeostasis, and show that high-resolution analysis of the GenERA dataset can be used to extract functional features of MREs. This study provides a genome editing-based multiplex strategy for direct functional interrogation of RNA *cis*-regulatory elements in a native cellular environment.

[1] Weatherall Institute of Molecular Medicine, Radcliffe Department of Medicine, University of Oxford, Oxford OX3 9DS, UK. [2] European Molecular Biology Laboratory-European Bioinformatics Institute, Wellcome Trust Genome Campus, Hinxton, Cambridge CB10 1SD, UK. [3] Center for Genome Engineering, Institute for Basic Science (IBS), Seoul 08826, Republic of Korea. [4] Molecular and Population Genetics Laboratory, Oxford Centre for Cancer Gene Research, Wellcome Trust Centre for Human Genetics, University of Oxford, Oxford OX3 7BN, UK. [5] Weatherall Institute of Molecular Medicine, Developmental Immunology, University of Oxford, Oxford OX3 9DS, UK. [6] Weatherall Institute of Molecular Medicine, MRC Molecular Haematology Unit, NIHR Oxford Biomedical Research Centre Programme, University of Oxford, Oxford OX3 9DS, UK. [7] Wellcome Trust Centre for Human Genetics, Oxford OX3 7BN, UK. [8] Department of Biomedicine, Laboratory of Paediatric Immunology, University of Basel, CH-4058 Basel, Switzerland. [9] Department of Chemistry, Seoul National University, Seoul, 151-747, Republic of Korea. [10] Sir William Dunn School of Pathology, University of Oxford, South Parks Road, Oxford OX1 3RF, UK. [11] Present address: Wellcome Trust Sanger Institute, Wellcome Genome Campus, Hinxton, Cambridge CB10 1SA, UK. Qianxin Wu and Quentin R.V. Ferry contributed equally to this work. Andrew R. Bassett and Tudor A. Fulga jointly supervised this work. Correspondence and requests for materials should be addressed to T.A.F. (email: tudor.fulga@imm.ox.ac.uk)

During cellular differentiation and function, the rate of protein production and turnover are precisely regulated at multiple levels. Within this context, intragenic non-coding RNA cis-regulatory elements (RREs) provide an essential post-transcriptional control layer, through regulation of RNA stability, localisation and processing[1]. RREs are defined by characteristic sequence motifs that serve as docking sites for trans-acting factors, such as short non-coding regulatory RNAs (e.g. microRNAs) and RNA binding proteins (RBPs)[2]. Although this knowledge enabled the development of in silico tools for genome-wide identification of putative RREs, studying the regulatory impact of these elements in an endogenous cellular context has been challenging[3].

Traditionally, studies aiming to decipher the functional role of RREs have relied primarily on fusing RRE-containing UTRs (or a region flanking the predicted element) to reporter genes, which provide an indirect readout of their activity[4–6]. Notably, such constructs are often overexpressed and inherently remove the RREs from their native RNA context. This is an important consideration since the susceptibility to RRE-mediated regulation is influenced by a variety of factors including local sequence environment, transcript abundance, effector level, cell state and the interplay between multiple cis and trans elements[7, 8]. Therefore, extrapolating the physiological relevance of putative RREs from artificial assays that do not recapitulate endogenous conditions is inherently difficult and potentially misleading. Consequently, although a vast spectrum of RRE classes has been identified[9], their cumulative and cooperative effects on the half-life of individual transcripts, as well as their broad impact on transcriptome homeostasis remain poorly understood[2, 7]. Therefore, the development of technologies enabling precise disruption of RRE motifs and direct quantification of the ensuing phenotypic changes at transcript level in a native cellular environment is of great importance.

microRNA (miRNA) response elements (MREs) are one of the most extensively characterised and abundant classes of destabilizing cis-RREs[9–12]. MREs are bound by miRNAs and mediate post-transcriptional regulation of gene expression primarily via a stepwise process ultimately leading to mRNA decay[13–17]. The major specificity determinant of MRE activity is a short sequence complementary to the 'seed' region comprising nucleotides 2–7 at the 5′ end of miRNAs[18, 19]. This knowledge has spearheaded the development of a large number of MRE prediction algorithms, which proved instrumental in guiding functional studies[20]. However, although the merit of these in silico tools for miRNA research is undeniable, their predictive power is limited by both false positive and false negative results[21].

To better understand MRE determinants in vivo, a range of crosslinking-based strategies has been developed for high-throughput experimental identification of miRNA-target binding events (Ago-HITS-CLIP, PAR-CLIP, iCLIP, CLASH)[12, 22–27]. Research employing these tools has revealed that miRNAs could efficiently bind transcripts bearing seed-containing canonical motifs, as well as a repertoire of atypical sites[12]. However, it has become apparent that not all binding events translate into functional target repression[28]. Therefore, assessing the regulatory significance of MRE networks under endogenous conditions remains one of the most important yet unmet technical challenges in this field.

The clustered regularly interspaced short palindromic repeats (CRISPR)-Cas9 system has emerged as a powerful tool for mutagenesis studies[29]. Here we have developed genome-engineering based evaluation of RNA regulatory element activity (GenERA), a CRISPR-based technology platform that enables unbiased screening for regulatory elements encoded in untranslated regions (UTRs) and direct functional analysis of RREs in a native cellular context. This strategy relies on coupling highly efficient NHEJ-based mutagenesis of discrete genomic loci with parallel quantification of the impact of each mutation on transcript abundance by next generation sequencing (NGS). The endogenous activity of any cis-RRE motif can thus be directly measured as a function of phenotypic variations in gene expression levels resulting from ablating its sequence.

To establish the GenERA pipeline, we initially designed a custom Streptococcus pyogenes Cas9 (SpCas9) single guide RNA (sgRNA) tiling library and carried out the first unbiased functional dissection of the entire post-transcriptional regulatory landscape encoded within a candidate 3′UTR. This analysis revealed that only deletions spanning a defined segment of this 3′ UTR caused measurable changes in transcript levels, suggesting that most regulatory elements were clustered within this region. Intriguingly however, an in silico prediction uncovered a relatively broad distribution of putative miRNA response elements (MREs) across this ~ 400 bp 3′UTR. To gain further insight into the post-transcriptional regulation of this gene, we next targeted in a multiplex fashion all predicted MREs and assessed their individual and combinatorial impact on steady-state gene expression output. Using this approach, we show that MREs displaying marginal regulatory activity in isolation can elicit strong cooperative destabilizing effects on transcript levels when acting in combination with other distantly located elements. Finally, we apply GenERA to interrogate the broad MRE network of a highly conserved miRNA (miR-184) in its native cellular context. This analysis revealed that although most predicted canonical MREs are functional, the potency of their activity varies significantly. We also demonstrate that this data can be used to dissect the sequence determinants underlying MRE functionality, and recapitulate the importance of seed nucleotides in miRNA activity. This study provides a CRISPR-based technology platform for unbiased discovery and functional characterization of intragenic RNA cis-regulatory elements in an endogenous cellular context.

## Results

**Analysis of native RRE activity by CRISPR-based mutagenesis.** We have previously shown that homology directed repair (HDR) could be used to mutate MREs and assess their effect on mRNA levels in a mixed cell population[30]. Although informative, this strategy cannot be easily scaled up for the analysis of large numbers of RREs or unbiased screens of intragenic non-coding regions. We reasoned that coupling efficient induction of genomic deletions by error prone NHEJ with next generation sequencing (NGS) as readout of activity, could alleviate this limitation enabling rapid and multiplexed analysis of RRE functionality under normal cellular homeostasis.

To assess the feasibility of this approach, we tested the activity of known RREs in the 3′UTR of the Drosophila pck gene (Fig. 1a). Three SpCas9 sgRNAs were designed to target a control region (devoid of any annotated regulatory sequences), a previously validated destabilising RRE (miR-184 MRE)[30–32], and a stabilising RRE (polyadenylation (polyA) signal) (Fig. 1a). Following individual sgRNA transfections, NHEJ-based mutagenesis of each genomic locus was assessed by targeted high-throughput sequencing of genomic DNA (gDNA) and complementary DNA (cDNA). The impact of RRE ablation on pck mRNA levels was then measured by comparing the frequency of reads with deletions overlapping each relevant element in gDNA and cDNA sequencing libraries.

Assuming that miRNAs act primarily to decrease target stability, successful deletion of MREs should render an increase in transcript abundance. As expected, deletions overlapping the

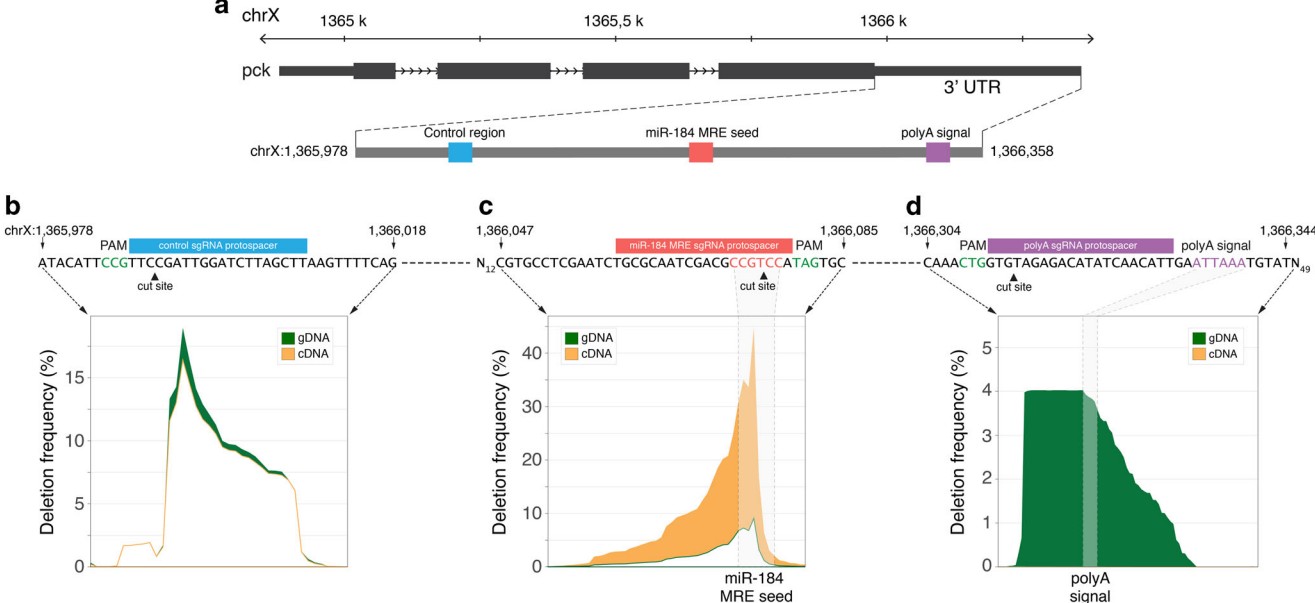

**Fig. 1** Experimental proof of concept for GenERA analysis. **a** *pck* genomic locus showing the 3′UTR relative coordinates of a control region, the predicted miR-184 MRE and the polyA signal. **b–d** Analysis of CRISPR-based mutagenesis results at each target region described in **a**. For each genomic locus, the identity of the PAM (*green*), Cas9-mediated DNA double stranded cut site (*arrow head*) and sgRNA protospacer target sequence (*blue*, *red* and *purple boxes*) are shown along the corresponding sequences within the *pck* 3′UTR. Differential analysis of nucleotide deletion profiles reflects negligible differences in cDNA/gDNA mutant read frequencies at the control region (**b**), a substantial enrichment of cDNA sequencing reads containing deletions in the miR-184 MRE seed (**c**), and a complete absence of cDNA reads with missing polyA signals (**d**). The percentage of deleted reads in cDNA and gDNA are shown in *orange* and *green* respectively; the position of predicted miR-184 MRE seed sequence and polyA signal are highlighted by *shaded* areas

control region had no apparent effect on the frequency of cDNA relative to gDNA mutant reads (Fig. 1b). However, disruption of the *pck* miR-184 MRE resulted in a significant overrepresentation of sequencing reads containing MRE deletions in cDNA compared to gDNA libraries (Fig. 1c). In contrast, ablation of the predicted polyA stabilising element rendered a complete loss of transcripts lacking this RRE, despite detection of many deletions covering this region in gDNA (Fig. 1d). These results demonstrate the potential of employing CRISPR-based mutagenesis for uncovering the regulatory properties of putative RREs encoded in endogenous untranslated regions.

**Design and implementation of the GenERA platform**. Conceptually, in a GenERA-based screen, a dedicated CRISPR sgRNA library is first used to mutagenize a region of interest, each discrete deletion is then precisely mapped to the genome, and its effect on transcript levels is measured by NGS using a dedicated computational pipeline (Fig. 2a). We reasoned that this relatively simple experimental framework should enable a wide range of analyses, including unbiased mutagenesis of UTRs as well as targeted alteration of discrete RRE genomic loci. In this study, *Sp*Cas9 sgRNAs were designed using a custom algorithm[30] and cloned into a bicistronic Cas9:sgRNA-scaffold expression vector (pAc-Cas9-sgRNA[33]). The ensuing plasmid-based sgRNA libraries were delivered to *Drosophila* S2R + cells in an arrayed format. Following antibiotic selection, gDNA and total RNA were simultaneously extracted from each cell pool and gene-specific primers were used to generate targeted NGS libraries from both gDNA and cDNA. To reduce PCR bias resulting from amplification of multivariate DNA templates, each reaction was performed at optimal non-saturating cycling conditions in five independent replicates, which were subsequently pooled together

for NGS. Finally, regional or RRE specific scores were calculated reflecting the impact of deletions on mRNA levels.

Deciphering the activity of RREs using the GenERA pipeline requires an in depth comparative analysis of transcript level variations over a large repertoire of NHEJ-induced deletions paired between gDNA and cDNA libraries. Accordingly, we have developed a computational pipeline to characterise 'unique deletion patterns' (UDPs) across a region of interest (ROI) defined as the minimal genomic window containing all UDPs (Supplementary Fig. 1, see Methods). Analysis within ROIs enabled maximum recovery of sequencing reads, including those containing spurious PCR and sequencing errors outside the Cas9-edited region. To measure the effect of every deletion, a UDP normalized score (UNS) was computed by dividing the cDNA to gDNA read counts of the corresponding UDP and calibrating this value to the wild type cDNA/gDNA ratio (Supplementary Fig. 1). Finally, regional and RRE specific regulatory scores were calculated by averaging UNS values across groups of relevant UDPs. This enabled the identification of stabilizing (UNS < 1) and destabilizing (UNS > 1) effects on transcript levels, as well as a direct comparison between sets of UDPs containing discrete RRE deletions and those ablating adjacent, presumed inactive sequences. To estimate the dynamic range and detection sensitivity threshold of this method, we processed in parallel a library of barcoded RP49 amplicon serial dilutions. This analysis revealed a near-perfect correlation between the theoretical copy numbers and experimental read counts ($R^2 = 0.94$), suggesting that as low as 10 reads could be reliability and accurately quantified in this pipeline (Supplementary Fig. 2).

**Unbiased interrogation of a 3′UTR regulatory landscape**. Using the GenERA platform, we first sought to survey the entire post-

transcriptional RNA regulatory landscape of a candidate 3′UTR in S2R+ cells. *CG9257* is a gene robustly expressed in this cell line and encodes a 395 bp long 3′UTR making it ideally suited for direct analysis by next generation sequencing (Supplementary Note 1). Furthermore, among genes with a UTR length of 300–500 bp, the *CG9257* UTR is strongly enriched in predicted MREs for S2R+ expressed miRNAs suggesting that it could be subjected to post-transcriptional miRNA-mediated regulation (Supplementary Fig. 3). To exhaustively investigate the regulatory potential of the *CG9257* 3′UTR and uncover the distribution of putative active zones, we identified all NGG and NAG protospacer adjacent motifs (PAMs) within this sequence and designed a corresponding saturating *Sp*Cas9 sgRNA library (Fig. 2b, Supplementary Data 1, Supplementary Note 1). This library consisted of 42 sgRNAs tiling across the entire length of the UTR with a median of 5 bp between consecutive Cas9 cleavage sites. Since the empirically determined mean NHEJ-mediated deletion size in S2R+ cells is ~ 20 nt[33], this distribution should render a complete deletion coverage overlapping nearly every UTR nucleotide. Only the most distal 25 nucleotides (nt), which included the poly A

signal, contained no predicted PAM sequences due to a very high percentage of A and T nucleotides (74%). To prevent the formation of large deletions generated by simultaneous genomic cleavage events, each sgRNA containing plasmid was delivered to cells individually, in an arrayed format, and all library samples were subsequently pooled after mutagenesis and antibiotic selection. Following targeted NGS of amplicons spanning the entire *CG9257* 3′UTR (Supplementary Data 2), gDNA and cDNA reads were mapped to the reference sequence, and all UDPs were derived using the GenERA pipeline.

Evaluation of this dataset revealed that ~ 93% of the UTR was covered by at least one UDP with a mean value of 12.8 UDPs per nucleotide position and a median UDP length of 19 bp (Fig. 2b, c). Comparative analysis of cDNA and gDNA deletion frequencies covering each base pair showed a substantial increase in cDNA reads in the region proximal to the STOP codon (Fig. 3a, Zone A, nucleotides 1–139). This effect suggests the presence of destabilizing regulatory elements within this part of the UTR. In contrast, deletions mapping to the rest of the UTR (Fig. 3a, Zone B, nucleotides 140–395) displayed very modest differences

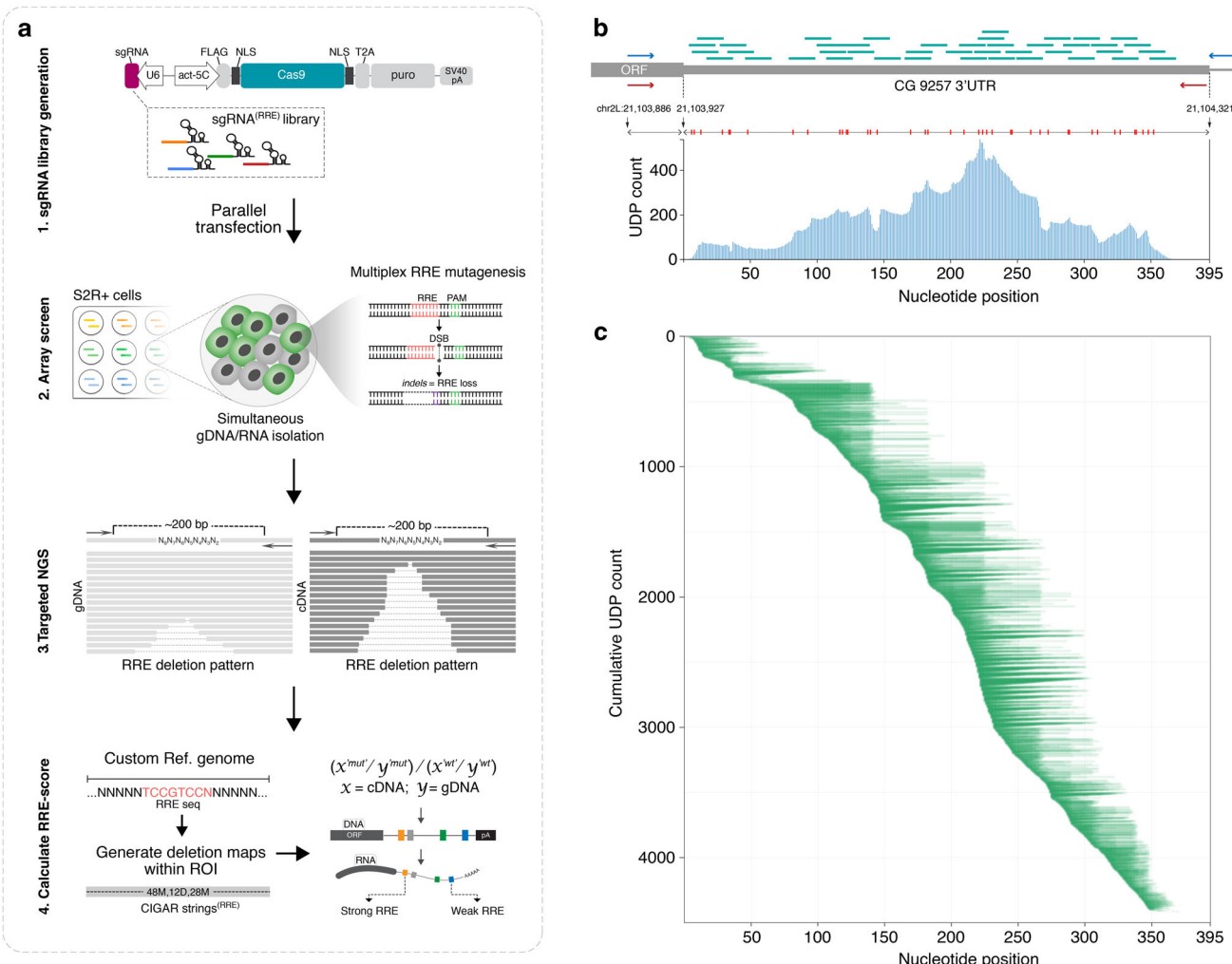

**Fig. 2** GenERA-based high-content mutagenesis of a candidate 3′UTR. **a** Design and implementation of experimental steps underlying GenERA-based parallel functional interrogation of RNA regulatory elements. **b** Genomic coordinates of *CG9257* 3′UTR showing the position and distribution of protospacers corresponding to all sgRNAs used to target this region (*green*). *Red bars* show the position of *Sp*Cas9 cut site for each sgRNA in the library. The relative locations of gDNA and cDNA NGS library primers are indicated by *blue* and *red arrows* respectively. *Blue histogram* reflects the total number of UDPs covering each individual nucleotide across the targeted region. **c** Coverage and distribution of all UDPs sorted by the position of the first deleted nucleotide and length of deletion. The position of nucleotides across the *CG9257* 3′UTR is shown on the x axis and the cumulative unique deletions count on the y axis

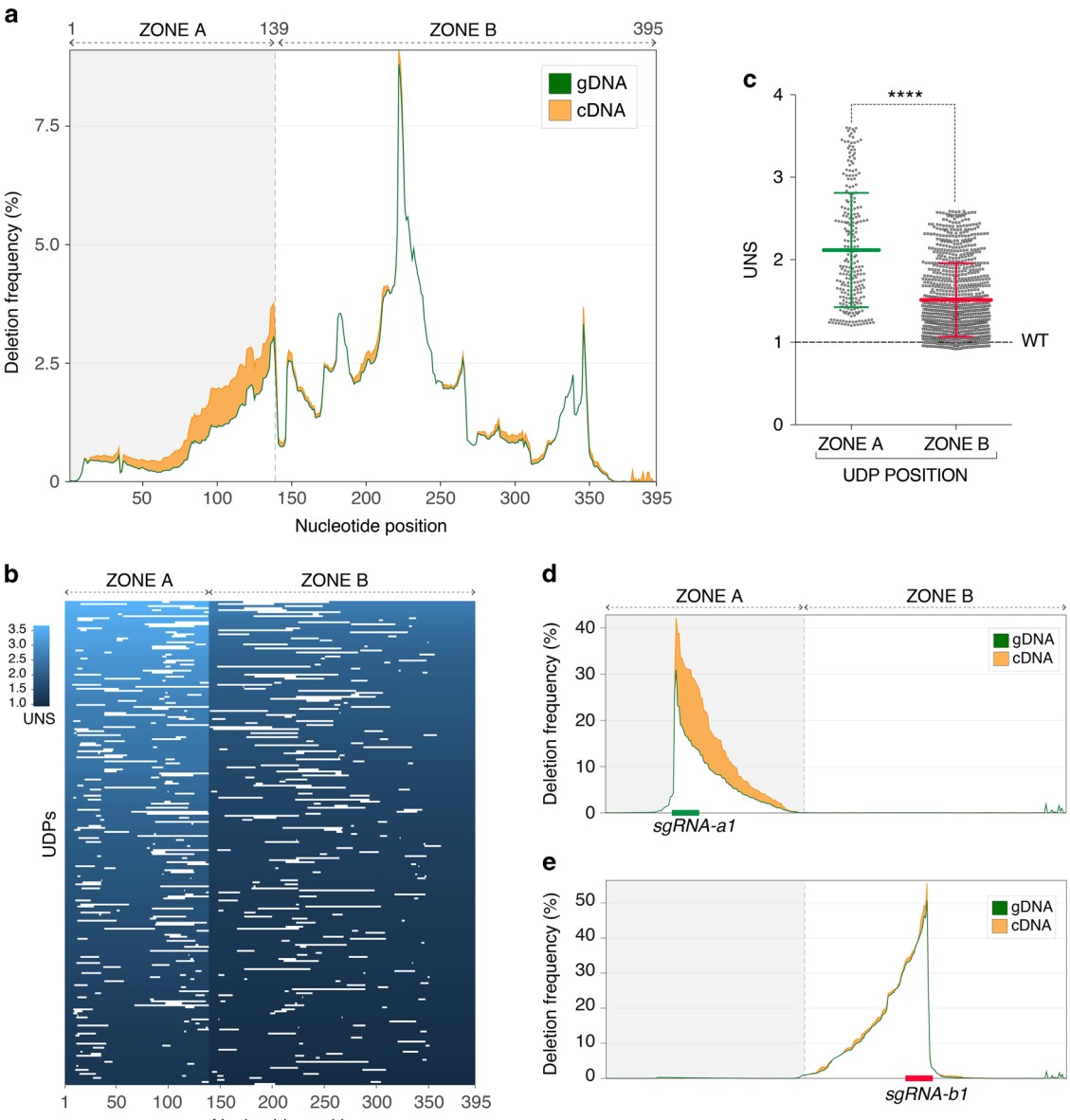

**Fig. 3** Unbiased surveillance of 3'UTR *cis*-regulatory potential with GenERA. **a** Analysis of nucleotide deletion frequencies in cDNA (*orange*) and gDNA (*green*) across the *CG9257* 3'UTR shows robust regulatory activity in region proximal to the open reading frame (ORF) (Zone A) and marginal activity in the rest of the UTR (Zone B). **b** Distribution of Zone A and Zone B UDPs (*white lines*) and their corresponding UNS values (*blue gradient*). The same number of UDPs was randomly sampled for both zones. **c** Comparative analysis of all UNS values (first to last quartiles) reflects significantly higher destabilising regulatory activity in Zone A compared to Zone B ($n = 259$ for zone A, $n = 1003$ for zone B, error bars = mean+/− SD, Mann-Whitney test, ****$P < 0.0001$). **d**, **e** Validation of observed differential regulatory potential associated with Zone A and Zone B. Peaks represent gDNA (*green*) and cDNA (orange) nucleotide deletion frequencies generated using individual sgRNAs designed to target Zone A (**d**) and Zone B (**e**). The precise positions of the sgRNA protospacers for Zone A (*sgRNA-a1*, *green*) and Zone B (*sgRNA-b1*, *red*) are displayed on the x axis

in cDNA/gDNA deletion frequencies, suggesting that, in isolation, this relatively extended region has only a marginal impact on transcript levels. Of note, the polyA signal which is located at the most distal end of Zone B and is expected to play a very important role in maintaining transcript stability and nuclear export, was not deleted in this instance due to technical constrains (see above).

To validate these results, we then calculated the corresponding UNS for each individual UDP across the UTR. This rigorous characterization of the dataset confirmed a bimodal distribution with UDPs mapping to Zone A consistently showing higher UNS values compared to Zone B (Fig. 3b, c). Finally, to

corroborate these findings, we delivered unique sgRNAs that specifically targeted Zone A or Zone B and assessed the effects of deleting nucleotides within these regions in isolation (Supplementary Data 1). Consistent with the results from the unbiased screen, reads containing deletions within Zone A were reproducibly enriched in cDNA relative to gDNA (Fig. 3d, Supplementary Fig. 4a), while deletions covering Zone B showed only marginal differences (Fig. 3e, Supplementary Fig. 4b). These results suggest that GenERA can be used in an unbiased manner for surveying the regulatory potential encoded within 3'UTRs and for guiding the discovery and physical mapping of active *cis*-RREs.

**Analysis of RRE combinatorial effects by multiplex GenERA.** The unbiased interrogation of *CG9257* 3′UTR suggested that most of the regulatory activity is clustered within a region proximal to the STOP codon (Zone A), and that the underlying RREs have primarily a destabilising effect on transcript levels. Since this behaviour could be indicative of miRNA-mediated

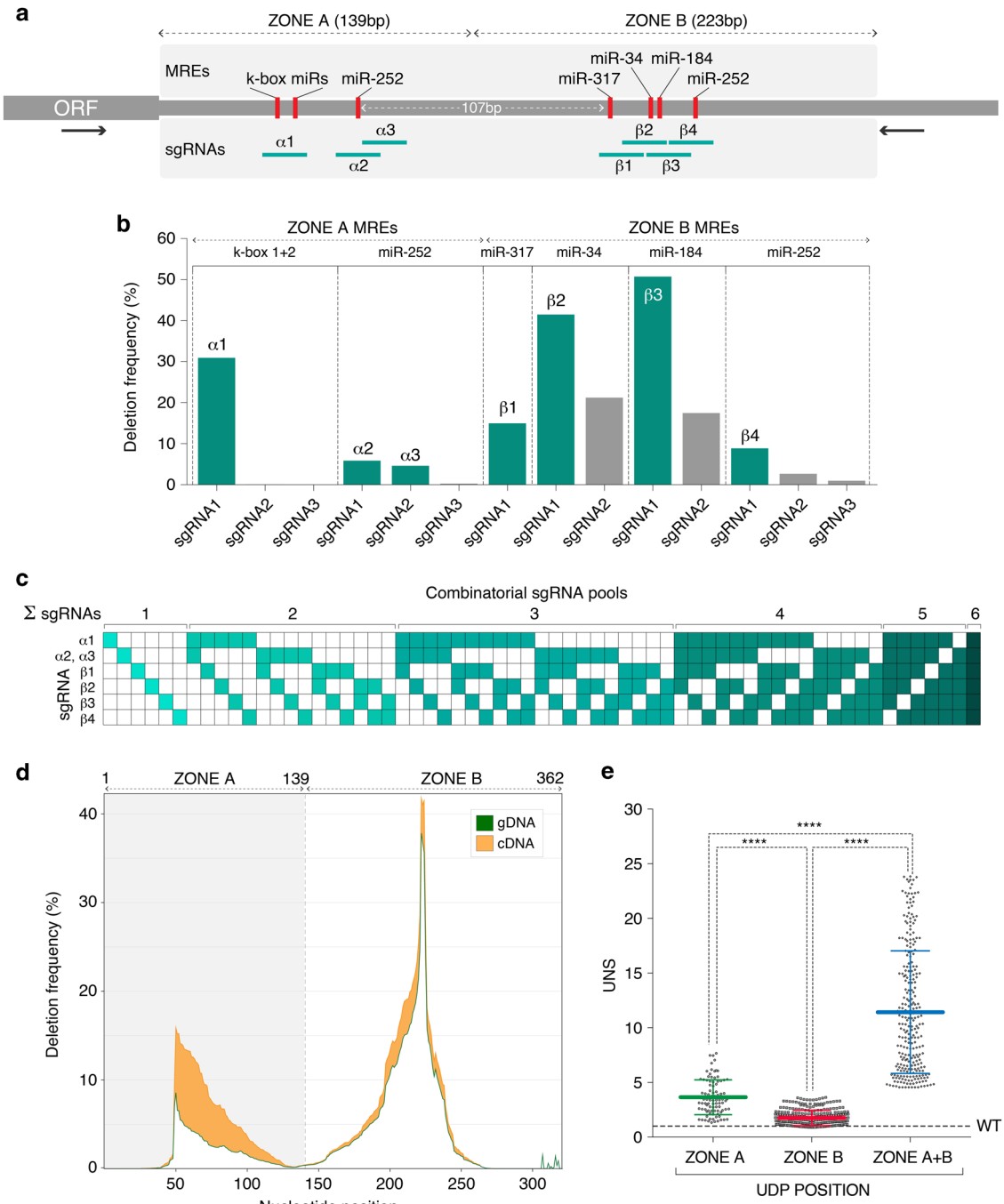

**Fig. 4** Implementation of GenERA for combinatorial RRE analysis. **a** *CG9257* 3′UTR displaying the boundaries of Zone A and Zone B, identity and position of all predicted MREs (*red*; low stringency *miRanda* target prediction algorithm), final sgRNAs designed to target each MRE (*green*), and gDNA/cDNA NGS library primers (*black arrows*). **b** The efficiency of all sgRNAs was tested by NGS and represented as percentage of reads containing deletions in the gDNA library (y axis). Final sgRNAs (*green*) were selected based on their efficiency and position relative to the seven predicted MREs. Since sgRNAs α2 and α3 which targeted zone A miR-252 MRE had relatively low efficiencies (5.8 and 4.6% respectively), they were delivered together in all combinatorial pools in order to increase the chance of generating miR-252 MRE deletions. **c** sgRNA multiplex strategy. All possible individual and combinatorial sgRNA pools (*n* = 63) were delivered to cells in an arrayed format. *Green squares* illustrate sgRNA identity in each given pool. Since sgRNAs α2 and α3 only targeted one MRE (Zone A miR-252) and had relatively low efficiencies (**a**), they were delivered together in all combinatorial pools in order to increase the chance of generating miR-252 MRE deletions. **d** Analysis of nucleotide deletion frequencies in cDNA (*orange*) and gDNA (*green*) generated by all combinatorial sgRNA pools in **c** reveals the regulatory activities associated with Zone A and Zone B. **e** Distribution of UNS values (first to last quartiles) calculated for all UDPs that overlap with Zone A (*green*), Zone B (*red*) and those concomitantly associated with Zone A and B (*blue*) (*n* = 93 for Zone A, *n* = 300 for Zone B, *n* = 272 for Zone A + B, error bars = +/− SD, Mann-Whitney test, ****$P$ < 0.0001)

regulation, we used the *miRanda* algorithm (lowest stringency) to map on this UTR the position of all predicted MREs targeted by the 10 most abundantly expressed miRNAs in this cell line[34] (Supplementary Data 3). Indeed, a cluster of three putative MREs mapped to the centre of Zone A, of which two belonged to the k-box miR family and one to miR-252 (Fig. 4a). Surprisingly however, this analysis also uncovered an additional cluster of four MREs (targets of miR-317, miR-34, miR-184 and miR-252, respectively) in a region of the UTR that appeared to only display modest regulatory potential (Zone B) (Fig. 4a). Assuming that all these miRNAs are active in S2R+ cells and capable of targeting

this transcript, at least two plausible scenarios could reconcile these findings. (i) Local contextual features, such as RNA secondary structure[31] or protein occupancy[7], dampen the repressive activity of MREs located in Zone B. (ii) These MREs act in concert with other RREs, in which case their isolated ablation in the unbiased GenERA screen would not necessarily cause a significant increase in transcript levels.

In principle, GenERA can be used to study the native functional relevance of any number of candidate RREs, and multiplexed to uncover combinatorial effects between distantly spaced elements within the same UTR. Accordingly, we set out to

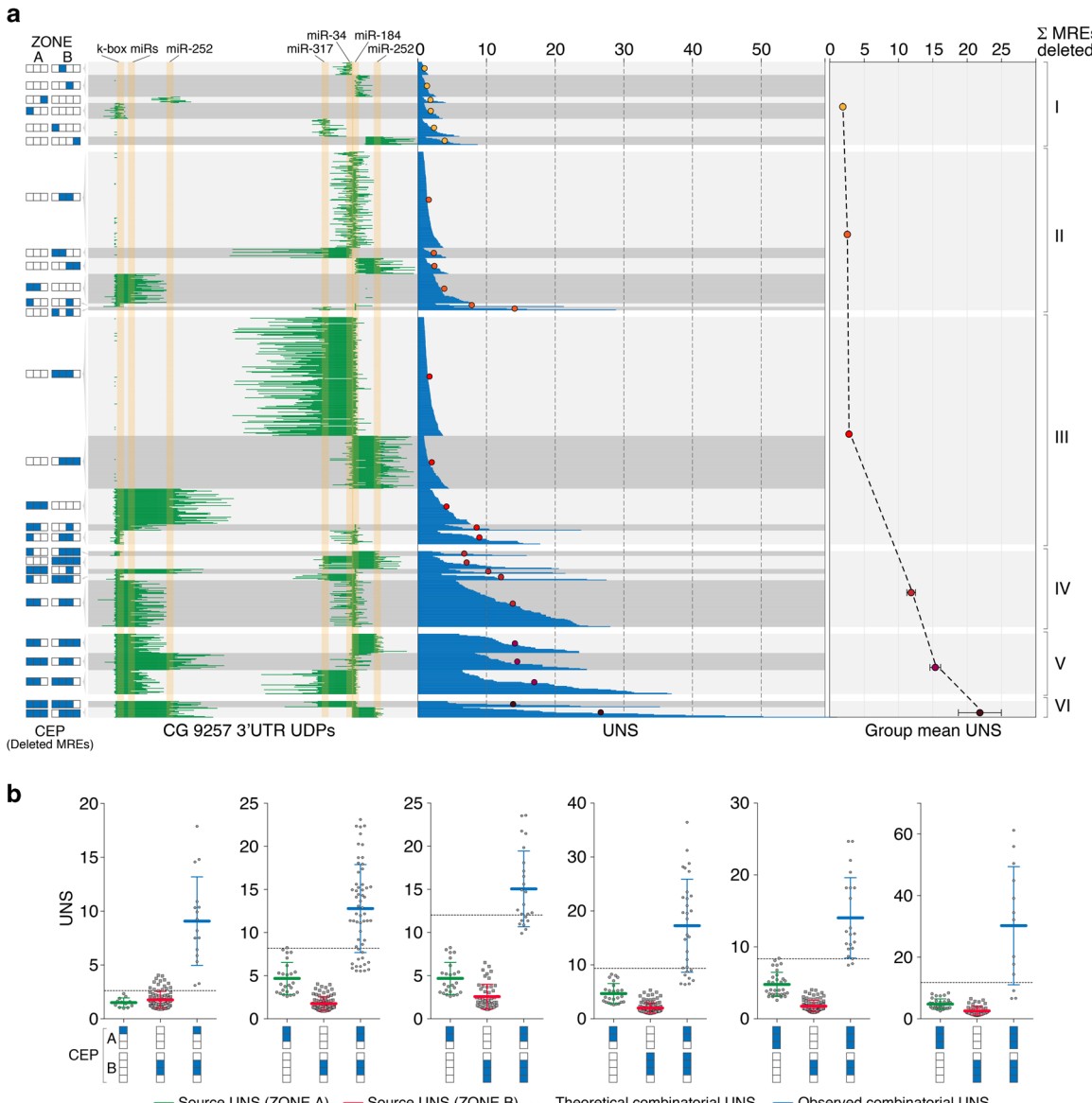

**Fig. 5** Multiplex GenERA uncovers cooperative regulatory activities between putative MREs. **a** Summary of MRE combinatorial UDP analysis. The *blue/white box* code on the far left displays all combinatorial editing patterns (CEPs) that passed a minimum UDP count filter (> 10 UDPs) and were included in the analysis (*blue* = MRE deleted; *white* = MRE intact). *Green lines* reflect the identity of each UDP associated with a corresponding CEP. *Yellow shaded* areas indicate the position of the seven predicted MRE seed sequences. UNS values were calculated for each UDP (*blue histogram*, first to last quartiles) and used to derive an average UNS value for each CEP (*coloured dots*). All CEPs were finally clustered based on the number of total deleted MREs (1–6) and a mean UNS value was calculated for each group (*right* panel, error bars = SEM). **b** Analysis of MRE cooperative activity. Scatter plots represent UNS values (first to last quartiles) of different Zone A (*green*) or Zone B (*red*) source CEPs, their corresponding theoretical combinatorial UNS (*dashed line*), and the observed combinatorial UNS (*blue*). Each individual datapoint represents an experimentally determined UNS value. In all cases analysed, the observed combinatorial UNSs are significantly higher than the predicted values suggesting the presence of cooperative interactions between Zone A and Zone B MREs (Error bars = +/− SD, Mann-Whitney test, ****$P < 0.0001$)

dissect the repressive potential of the MREs encoded in the *CG9257* 3′UTR and elucidate the overall impact of miRNA-mediated regulation on the steady-state transcript levels of this gene. First, we designed a panel of 14 sgRNAs and selected seven to individually target the discrete genomic loci encoding each predicted MRE in zone A and B of *CG9257* 3′UTR (Fig. 4a, b, Supplementary Data 1, Supplementary Note 2). The number of sgRNAs per MRE was dictated by the proximity between two adjacent MREs, PAM availability and the empirically determined sgRNA efficiency (Fig. 4b and Supplementary Note 2). To investigate the possibility of multiple MREs acting in concert to control *CG9257* expression, we then multiplexed all MRE-specific sgRNAs to obtain the maximum number of combinatorial editing patterns (63 combinatorial pools in total, Fig. 4c). The resulting library was again delivered to S2R+ cells in an arrayed format. Following antibiotic selection, all samples were harvested together and pooled prior to NGS analysis.

We first set out to determine whether the results obtained in the unbiased mutagenesis screen could be recapitulated using this new set of combinatorial sgRNAs. Consistent with previous data, an aggregate analysis of all deletion frequencies revealed a marked increase in cDNA reads overlapping Zone A, suggesting that the underlying MREs could impart strong repressive action (Fig. 4d). Interestingly however, in this case the frequency of deletion-containing reads was also increased to some extent in cDNA relative to gDNA libraries in Zone B, albeit to a lower degree than those in Zone A (Fig. 4d). This result could be explained by the presence of discrete combinatorial deletions simultaneously mapping to both zones. To test this possibility, we demultiplexed all UDPs into those mapping exclusively to Zone A, B or Zone A + B concomitantly, and calculated their corresponding UNS values (Fig. 4e). As expected, this analysis confirmed significantly higher regulatory activity associated with Zone A compared to Zone B. Importantly, and consistent with our hypothesis, discontinuous UDPs mapping to Zone A + B simultaneously displayed significantly increased destabilizing activity (mean UNS > 10, Fig. 4e), suggesting possible cooperative effects between these MRE clusters.

**Dissection of cooperative miRNA-mediated regulation with GenERA.** Previous elegant studies employing a variety of reporter assays and computational analysis of transcriptome-wide profiling data, indicated that MREs located in close proximity on the same 3′UTR can act in a cooperative manner under certain circumstances[11, 35, 36]. To determine if GenERA could be used to study this phenomenon in a native cellular context, we first sorted all UDPs into combinatorial editing pattern (CEP) groups, representing identical combinations of single or multiple MREs deleted (Fig. 5a, left panel). We then calculated UNS values within each CEP group and plotted the distribution of these values relative to the cumulative number of deleted MREs (Fig. 5a, middle panel). This analysis revealed that in general, CEPs containing single MRE deletions show weaker destabilising activity compared to those affecting multiple MREs, and the potency of repression increases with CEP complexity (Fig. 5a, right panel). The data also suggests that the paired k-box MREs located in Zone A impart considerable repression in isolation, and more prominently when acting in concert with other Zone A or Zone B MREs. However, due to the complexity of the deletion repertoire generated here it is difficult to unequivocally derive the potency of individual MREs from this analysis.

To determine whether MREs located in Zone A and B might act in a cooperative manner to regulate *CG9257* expression, we isolated sets of CEPs which contained MRE deletions in Zone A or Zone B only (source CEPs) and had corresponding matching deletions in both zones (combinatorial CEP) (Fig. 5b). To increase

confidence, we included in the analysis only CEPs which contained > 10UDPs in all three conditions. Comparing the activity (UNS values) of source CEPs confirmed the dominant effect of Zone A MREs, and highlighted the potency of the second k-box MRE located in this region. Assuming that no functional relationship exists between Zone A and Zone B MREs, the observed activity of combinatorial CEPs should match the additive score of individual source CEPs[11]. To determine this value we calculated a theoretical combinatorial UNS based on multiplying the UNS means (reflecting fold change derepression) of corresponding source CEPs (Fig. 5b, see Methods). Interestingly, for all CEP groups included in this analysis, the observed combinatorial UNS was markedly higher than that predicted by additive effects, strongly suggesting that MREs located in Zone A and B act in a cooperative manner to regulate *CG9256* transcript levels at steady state (Fig. 5b). These results demonstrate the utility of applying GenERA to uncover combinatorial and cooperative activities between multiple *cis*-RREs in a native cellular context.

**Functional interrogation of a native MRE network.** We next sought to establish the potential of GenERA for multiplex interrogation of MRE networks in a native cellular context. To carry out this analysis, we chose the predicted *Drosophila* miR-184 MRE network which is sufficiently complex to allow a comprehensive analysis, but at the same time amenable to an arrayed screening format. Furthermore, miR-184 is a highly conserved miRNA (through to vertebrates) with pleiotropic functions in development and disease, and one of the two most abundantly expressed miRNAs in *Drosophila* S2 cells[32, 34, 37–39]. Using two target prediction algorithms (*miRanda-mirSVR*[40] and *TargetScanFly 6.2*[34]) we identified conserved canonical miR-184 MREs encoded in *Drosophila* 3′UTRs (Supplementary Fig. 5a and Supplementary Data 4). In addition, all miR-184 MREs reported in two previous studies[31, 32] as well as predicted MREs in long intergenic non-coding RNAs (lincRNAs) were considered when building this network (Supplementary Fig. 5a). To reduce the probability of false negative events, an expression threshold filter based on S2R+ transcriptional profiling was then applied, to exclude genes that have no detectable transcripts in this cell line (Supplementary Fig. 5a and Supplementary Data 5). The resulting network consisted of 77 putative miR-184 targets representing all major classes of canonical MREs (8mer, 7mer-m8, 7mer-A1, 6mer and single G:U wobbled seed site) (Supplementary Fig. 5b, c and Supplementary Data 4). Since GenERA-based analysis relies on altering the MRE genomic loci, to account for possible miR-184-independent effects, we also annotated other predicted RREs within a 200 bp window flanking each putative MRE. These included MREs of other highly expressed miRNAs in S2 cells[34], AU-rich elements (AREs), polyadenylation signals (PAS), Pumilio response elements (PREs) and HuR binding motifs (HBMs) (Supplementary Fig. 6 and Supplementary Data 4 and 6).

To systematically edit each MRE within the native miR-184 network, we searched for *Sp*Cas9 NGG and NAG protospacer adjacent motifs (PAMs) proximal to all predicted miR-184 MRE seed sequences (Supplementary Note 3 and Supplementary Data 7). A GenERA single guide RNA (sgRNA) library was then generated and delivered to S2R+ cells in an arrayed format. Subsequently, genomic DNA (gDNA) and RNA were simultaneously extracted from each sample, individually amplified with gene specific primers flanking the miR-184 MRE regions, and all amplicons were pooled together for NGS library preparation. Following targeted sequencing, all gDNA and complementary DNA (cDNA) reads were mapped to a custom reference genome defined by a 200 bp window centred on the coordinates of each miR-184 MRE seed sequence (Supplementary Data 8). We found

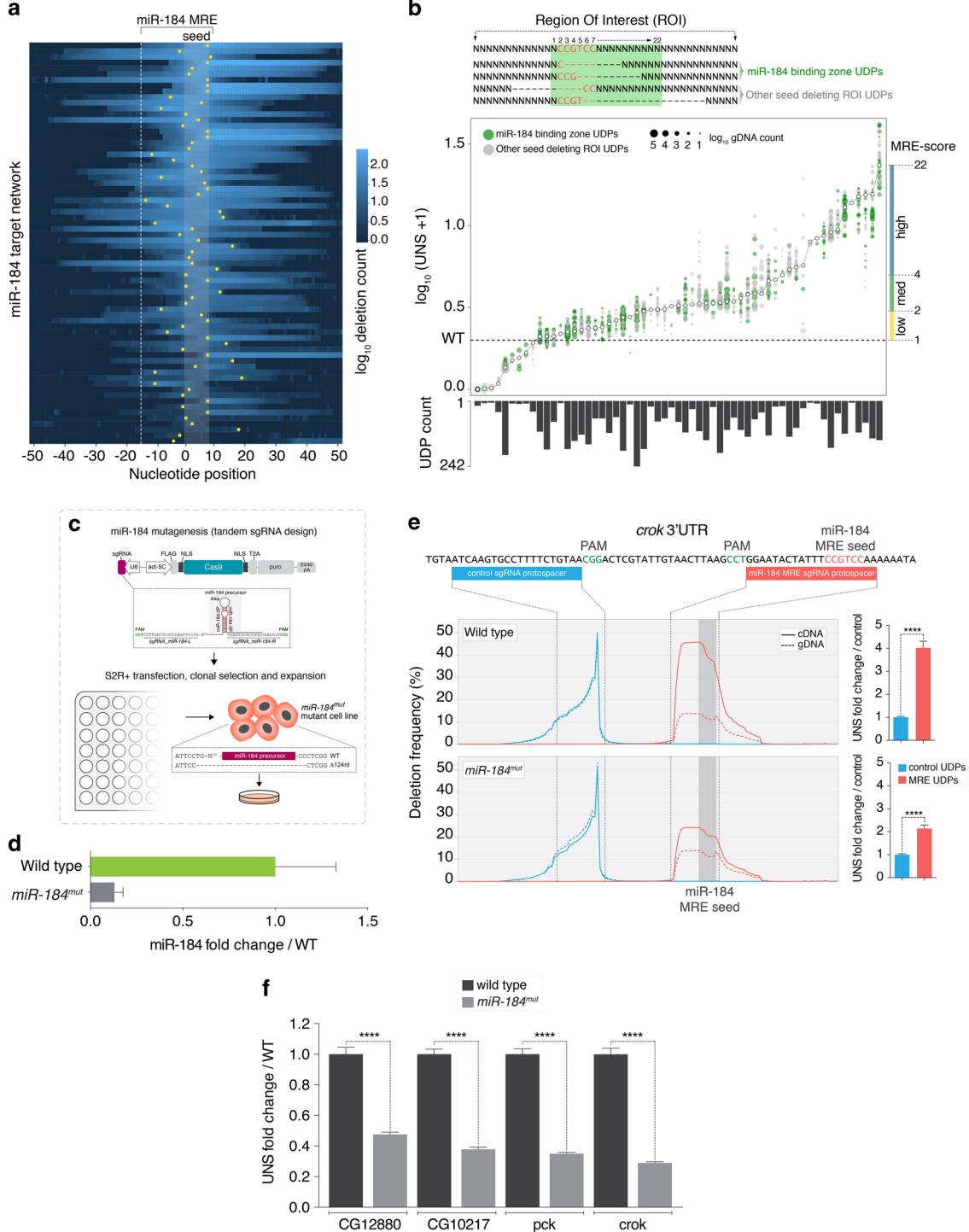

**Fig. 6** GenERA-based analysis of a predicted MRE network activity. **a** UDP repertoire (*blue gradient* = number of UDPs overlapping a given nucleotide) across the miR-184 target network. Boundaries of the predicted miR-184 binding zone (*dashed line*), extended seed region (*shaded* area), and Cas9 DNA double stranded break sites (*yellow dots*) are highlighted. **b** Spatial distribution of seed-deleting UDPs selected for MRE-score calculation (*top*). UDPs restricted to the miRNA binding zone (*green* nt. 1–22) and extending to the full ROI (*gray*) were considered in the analysis. Ranked MRE-score distribution across the miR-184 MRE network (*white dots*) (*bottom*). Underlying UNS values (first to last quartiles) are shown with their sequencing depth (dot size) and spatial distribution (*green* and *gray*). Bar plot reflects total UDP counts contributing to each MRE-score calculation. Right y axis shows partitions in *high*, *medium*, and *low* MRE-score groups based on the empirical distribution of the data. **c** Generation of *miR-184^mut* cell line by CRISPR-Cas9 genome editing. **d** Quantification of mature miR-184 levels by RT-qPCR in wild type cells and *miR-184^mut* cells (*n* = 3 for each group, error bar = SEM). **e** Validation of GenERA data. Analysis of *crok* cDNA and gDNA deletion frequency profiles in wild type and *miR-184^mut* cells at a control locus (*blue*) compared to the predicted miR-184 MRE locus (*red*) (left plots). Quantification of UNS fold change from MRE UDPs normalized to control UDPs in wild type and *miR-184^mut* cells (*right bar* graphs, Error bar = SEM, Mann-Whitney test, ****P < 0.0001). **f** Comparative GenERA analysis of four top-ranking miR-184 MREs reveals a significant fold change difference between wild type and *miR-184^mut* cells, demonstrating dependence of the observed effects on miR-184 mediated regulation (*Error bar* = SEM, Mann-Whitney test, ****P < 0.0001)

that 90.1% gDNA amplicons and 94.3% cDNA amplicons were sequenced at very high depth (Supplementary Fig. 7a). Reads carrying deletions were generally enriched in cDNA libraries relative to gDNA libraries (Supplementary Fig. 7b), a behaviour consistent with removing destabilizing elements (such as MREs). At the amplicon level, this phenomenon was observed in 58/70 genes across the miR-184 putative network (Supplementary Fig. 7c).

**Evaluation of miR-184 native MRE network activity with GenERA**. To decipher the native repressive activity of each MRE within the miR-184 target network, we characterised the set of genomic deletions created at every target locus using the GenERA computational pipeline. In total, 6913 UDP types were generated, with a median of nearly 90 per candidate MRE (Supplementary Fig. 8a). Three targets were excluded from the analysis on the account of no detectable seed deletions, lack of paired gDNA-cDNA deletions, and inefficient Cas9-mediated mutagenesis respectively (Supplementary Fig. 8b). Most sgRNAs in our library were designed to cut in close proximity or overlapping predicted miR-184 MRE seed sequences (Supplementary Data 7, Supplementary Note 3). As expected, a projection of all target-specific UDPs revealed that in nearly all cases the prevalence of UDPs was enriched across the MRE extended seed regions (nt. 1–8) (Fig. 6a). Overall, we found that the majority of MREs (87.5%) were covered by paired deletions of the entire extended-seed sequence (positions 1–8), with a further 8.3% containing partial seed deletions (Supplementary Fig. 8b). Interestingly, an assembly of all UDP pattern distributions revealed that irrespective to the genomic site, sgRNA directionality or UDP size, deletions almost exclusively extended away from the PAM sequence (Supplementary Fig. 9).

Because the MRE seed is a critical determinant of miRNA-mediated repression, only UDPs totally or partially overlapping with the extended seed sequence were considered when deriving a consensus effect of MRE ablation. In addition, a read count cut-off was also applied to select for high-confidence UDPs (see Methods). Since generation of *de novo* MREs resulting from genomic ablations could distort UNS values, all UDPs containing ectopic MREs targeted by the most abundant miRNAs in S2 cells (including *de novo* miR-184 MREs; Supplementary Data 3) were excluded from the analysis. In some instances, other putative RREs were found in very close proximity to the target MREs making it difficult to avoid their coincidental ablation. However, a rigorous evaluation of the UDP repertoire revealed that in most cases deletion of these putative RREs had minimal impact on UNS distribution (Supplementary Fig. 10). Since comparative analysis of all library replicates revealed a near perfect correlation of WT and UNS scores (Supplementary Fig. 11a, b) (average rectified $R^2 = 0.94$), these datasets were merged to further maximize UDP read counts and increase statistical power. Finally, all UNS values were compiled to calculate a weighted normalized MRE score (MRE-score; see Methods), reflecting the overall activity of each miRNA response element on transcript levels (Fig. 6b).

Analysis of MRE-scores revealed that altering the sequence of most MREs (~85%) comprising this canonical target network rendered a detectable increase in transcript abundance (MRE-score > 1), consistent with relief of miR-184-mediated repression (Fig. 6b and Supplementary Data 9). A certain degree of variability was observed between UNS values for a single MRE, presumably due to technical variations (sequencing depth, PCR amplification bias, etc) and/or differences in the seed deletion coverage and span of deletions. A small number of edited loci appeared to cause a decrease in MRE-score, suggesting the

presence of underlying RNA stabilizing elements. Although interesting, due to the underrepresentation of these instances in our dataset ($n = 9$), and the complexity of dissociating MRE-dependent effects from other coincidental RREs, these targets were excluded from further analysis.

Our results suggest that most MREs underlying this target network are functionally regulated by miR-184 under normal homeostasis. However, the MRE-score amplitude varied significantly within the MRE network, as well as between multiple MREs encoded within the same transcript (Fig. 6b and Supplementary Data 9). Approximately 60% of MREs appeared to dampen transcript abundance (1 < MRE-score < 4), while a smaller subset mediated robust repression (MRE-score > 4), potentially reflecting 'tuning' vs. 'switch' interactions respectively[41]. Validating the biological relevance of calculated MRE-scores, 95% of previously reported miR-184 targets appeared to be actively regulated in our analysis (MRE-score > 1) (Supplementary Data 9)[31, 32]. Finally, the repertoire of discrete UDPs also enabled an accurate dissection of tandem miR-184 MREs predicted in the 3′UTR of two genes, *sinu*[31, 32] and *CG31195*. In both cases, individual MREs displayed differential repressive activity (MRE-score $CG31195^{MRE1} = 4.4$, $CG31195^{MRE1} = 1.3$; MRE-score $sinu^{MRE1} = 2.4$, $sinu^{MRE2} = 1.8$), suggesting contextual differences between each target site.

**Validation of GenERA-based MRE network analysis**. To establish the reliability of this approach and confirm that the observed effects depend on mutating *bona fide* MREs, we sought to carry out comparable experiments upon manipulation of miR-184 levels. Using the *Sp*Cas9 nuclease and tandem sgRNAs, we generated a stable miR-184 mutant S2R+ cell line carrying a 124 nt genomic deletion encompassing the miR-184 precursor locus (*miR-184$^{mut}$*) (Fig. 6c). RT-qPCR analysis revealed 90% loss of mature miR-184 levels in this cell line (Fig. 6d). We initially compared the effect of sgRNAs targeting a top-ranking MRE candidate (*crok*) and an adjacent control region in wild type cells (Fig. 6e). As expected, deletions partially or fully covering the miR-184 MRE seed were enriched in cDNA, and caused a significantly higher UNS than at the control locus (Fig. 6e). Demonstrating the dependence of this effect on miR-184 activity, analysis of equivalent deletions in mutant cells exhibited a lower mean UNS fold change compared to WT cells. However, UNS values still displayed statistically significant differences from the control region, presumably due to residual miR-184 activity in *miR-184$^{mut}$* cells (Fig. 6d, e). Similar experiments were carried out on three other target genes from the top-ranking MRE-score group (*CG12880*, *CG10217* and *pck*) in wild type and *miR-184$^{mut}$* cells. All tested MREs reproducibly displayed robust destabilising effects, which were dependent on miR-184 activity as reflected by a significant reduction in UNS values in mutant vs. wild type cells (Fig. 6f).

We next sought to evaluate the correlation of MRE-scores with contextual features previously reported to influence the efficacy of miRNA-mediated repression[28, 42]. Based on the empirical distribution of the data, two inflection point boundaries were defined to partition all MREs into MRE-scores *low* (< 2), *medium* (2–4) and *high* (> 4) groups (Fig. 6b, Supplementary Fig. 12a and Supplementary Data 9). A set of candidate contextual features was assembled, which included MRE intrinsic factors (miR-184-MRE pairing thermodynamic stability, GC content, RNA accessibility, conservation) as well as features defining the MRE local environment (local AU content, MRE position, UTR length). We also tested other parameters such as *transcription activity* (estimated from a previous Pol-II ChIPseq dataset[43]) and the prevalence of other co-targeting miRNAs[44] (other than miR-184; *TargetScan 6.2*). Several features displayed the expected trend of correlation with the

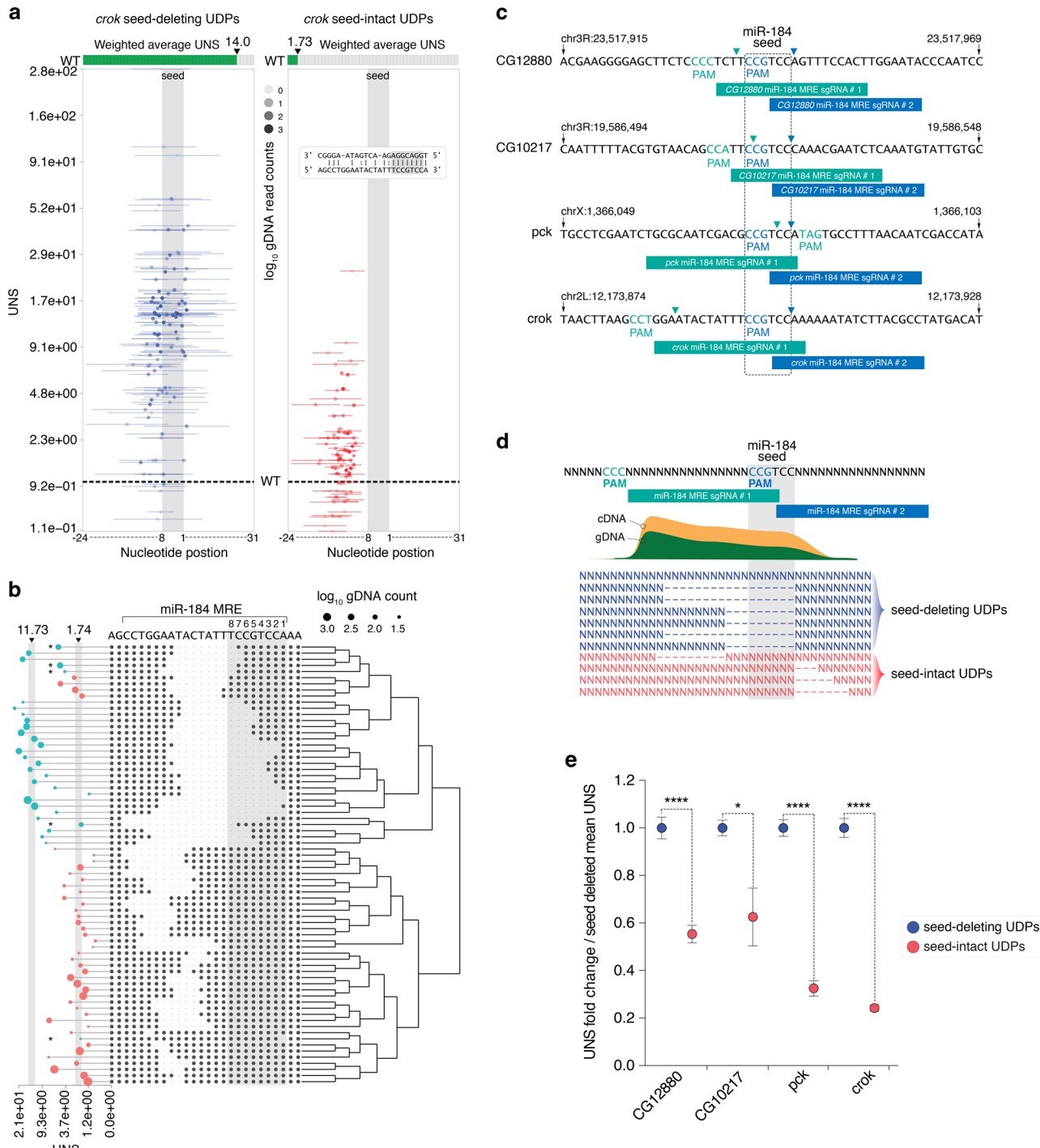

**Fig. 7** Functional dissection of MRE sequence determinants. **a** Mapping of the complete UDP repertoire across the *crok* ROI (beginning and end coordinates of the deletion peaks) shows deletions patterns (x axis, line = UDP span, dot = center), UNS (y axis) and sequencing depth (transparency). UDPs are partitioned relative to their coverage of the extended seed region (*blue* = seed deletion; *red* = seed intact). UNS values for the 'seed deletion' group are significantly higher than the 'seed intact' group (*t*-test, *P* < 0.0001). Top meter indicates average UNS values for each group weighted by UDP sequencing depth (*green*). **b** High resolution analysis of *crok* UDP sets restricted to the miR-184 MRE region (nt. 1–22). Dendrogram showing hierarchical clustering of UDPs based on their deletion footprint (*dashed line*) is displayed along with corresponding UNS values (circle and lines horizontal plot). Scores are scaled by sequencing depth (size of *circle*) and coloured depending on UDP coverage of the extended seed region (*teal* = seed deletion; *red orange* = seed intact; * = UDP affecting only seed distal nucleotides). **c–e** Analysis of sequence determinants underlying top ranking MREs from GenERA screen. To increase UDP complexity at each targeted locus, a second batch of sgRNAs was designed (*blue*) in addition to the ones used for the GenERA screen (*green*) (**c**). The identity of each protospacer (box), Cas9 cut site (arrow head) and PAM sequence are shown. **d** Diagrammatic representation of the computational approach used to demultiplex the ensuing UDP repertoire into seed-deleting (*dark blue*) and seed-intact (*red*) classes. **e** Seed-intact UDPs consistently generate lower UNS values compared to seed-deleting UDPs (Error bar = SEM, Mann-Whitney test, ****P* < 0.0001, **P* < 0.05)

three MRE-score groups, of which the most prominent one was the prevalence of co-targeting miRNAs (Supplementary Fig. 12b, c)[44]. This suggests that under normal cellular homeostasis and endogenous miRNA-target stoichiometry, some of these contextual features indeed play important roles in modulating MRE activity. Taken together, these results validate the relevance of GenERA analysis to the discovery of functional MREs.

**High-resolution analysis of MRE sequence determinants**. We reasoned that the diversity of Cas9-generated deletions overlapping individual miR-184 target sites could also enable high-resolution functional interrogation of MRE sequence determinants. To test the feasibility of this approach, we selected the unique miR-184 MRE encoded in the 3′UTR of *crooked* (*crok*), which appeared to display particularly strong repressive activity (Supplementary Data 9; MRE-score = 14.06). We first aligned all *crok* UDP footprints to the region around the MRE (x axis), and mapped their UNS (y axis) to generate an overview representation of the entire UDP set. These were further subdivided into seed-deleting and seed-intact sets according to their deletion patterns (Fig. 7a). We found that deletions covering the extended seed sequence (nt. 1–8) consistently generated high UNS values (blue, weighted mean UNS = 14.0), while UDPs landing outside of this region followed a distinctively separate distribution centred on the WT score value (red, weighted mean UNS = 1.73) (Fig. 7a). To more accurately resolve sequence determinants underlying miR-184-mediated repression, we next focused only on UDPs restricted to the 22 bp MRE segment predicted to interact with the mature miRNA. As anticipated, high UNS values were associated with deletions overlapping the *crok* seed sequence, highlighting this region as a critical determinant of miR-184-mediated repression (Fig. 7b; weighted mean UNS = 11.73). A careful examination of discrete *crok* deletions revealed that UDPs removing only nucleotide 8 of the extended MRE seed exhibited lower derepression potential, suggesting that base pairing at this position might not be essential for effective targeting (Fig. 7b). In addition, differential analysis of all seed-containing *crok* UDPs showed that deletions of core nucleotides (nt 2–7) consistently scored higher than those affecting only distal positions (nt. 8 or 1) (Supplementary Fig. 13). This observation is further supported by previous structural[45–47] and functional studies[11, 47–50].

To demonstrate the reliability of this approach, we extended the analysis to other top-ranking MREs from the GenERA screen in addition to *crok* (CG12880, CG10217, *pck*). First, to increase the complexity of seed-proximal UDPs we designed a second batch of sgRNAs targeting alternative sites near the predicted seed regions (Fig. 7c). sgRNAs from this batch and those from the initial GenERA screen were individually transfected, and cells were pooled together upon harvesting. Following NGS, all UDPs were computationally deconvoluted into seed deleting and non-seed deleting groups using the GenERA pipeline (Fig. 7d). Consistent with previous data, we found that for all tested targets, deletions covering the seed sequence generated significantly higher UNS values compared to those immediately outside of this region (Fig. 7e). Taken together, these results demonstrate that the diversity of GenERA-induced genomic deletions can be exploited to extract key sequence requirements of MRE function. In addition, this suggests that GenERA could be used to understand sequence determinants underlying the activity of other, less characterized, RREs.

## Discussion

Recent studies have revealed that during animal evolution, while the total number of protein coding genes has remained relatively constant, the median 3′UTR length has expanded from 130nt in worms to 780nt in humans[51]. In addition, the number of alternative 3′UTR isoforms per gene has also increased in higher organisms, especially in complex tissues such as the brain[51, 52]. These findings suggest that the impact of post-transcriptional control imparted by intragenic RNA *cis*-regulatory elements has significantly evolved with organismal complexity[53]. Indeed, a growing body of evidence suggests that UTRs play critical roles in regulating gene expression during development, and mutations abrogating the function of discrete RREs or loss of UTR sequences have been linked to various human diseases including cancer[54, 55].

While the importance of this regulatory layer is undeniable, the relatively limited availability of methods for functional analysis of RREs has so far restricted a comprehensive investigation of their function in an endogenous cellular context. Therefore, technology platforms enabling unbiased screening for regulatory activity encoded in UTRs, direct mapping of physiologically active RREs on native transcripts, and decoding the combinatorial impact of multiple RREs within the same UTR, are in great demand. An important advance in this direction came from the recent development of the RNA-element selection assay (RESA) a transcriptome-wide reporter system for investigation of RNA regulatory features[6]. Although this approach enables the analysis of RRE regulatory potential in vivo, it does not provide a direct evidence for their physiological relevance in a native context and under normal cellular homeostasis. The CRISPR-Cas9-based GenERA platform developed here enables direct functional interrogation and quantification of endogenous RRE activities by generating large repertoires of unique genomic deletions and measuring the frequency of their occurrence in the ensuing transcript pool. To our knowledge, GenERA represents the first account of a technology that enables direct, physical coupling of highly efficient NHEJ-based genome editing events to quantitative analysis of transcript copy number (i.e. phenotypic variations in gene expression levels) in a native cellular context.

We apply GenERA to survey in an unbiased manner the regulatory landscape of an entire candidate 3′UTR, and uncover the prevalence of putative active zones across its sequence. Interestingly, this exploratory approach suggested that the vast majority of regulatory potential is clustered at the beginning of the CG9257 3′UTR and this region encodes almost exclusively destabilizing elements. To gain further mechanistic insight into the post-transcriptional regulation of this gene, we used GenERA with multiplexed sgRNAs to deconstruct the combinatorial activity of all predicted MREs across its 3′UTR. Surprisingly, this in depth analysis uncovered much stronger silencing effects imparted by cooperative repressive action between distantly located MREs. This suggests that, in isolation, saturation mutagenesis studies may erroneously designate regions containing functional RREs working in concert with other distant elements, as devoid of regulatory potential. These results underscore the complexity of RRE-mediated regulation and highlight the importance of in depth combinatorial studies in elucidating the functional impact on gene expression of UTR-encoded elements.

We also use the GenERA platform to precisely and effectively assess the activity of each MRE node within a primary miRNA-target network under endogenous conditions and stoichiometry. We demonstrate that this experimental framework enables direct phenotypic evaluation of MRE activities in a multiplexed and semi-quantitative manner. Analysis of *Drosophila* miR-184 targets revealed that the vast majority of canonical MREs (85%) appeared to be actively regulated. However, across this network we observed a broad range of MRE activities from fine-tuning regulation to very strong repression. It should be noted however

that since a small subset of targeted MREs rendered a decrease rather than increase in transcript levels, we cannot exclude the possibility that some of the observed effects may be in part due to coincidental alteration of other, unidentified RREs. Although the GenERA experimental framework is ideally suited to evaluate the impact of MRE activity on transcript abundance, it is not designed to measure translational repression. At present however, the most prevailing miRNA-mediated silencing paradigm supported by robust experimental evidence, is a stepwise dynamic repression mechanism. According to this model, miRNA binding initially triggers translational interference, which is almost invariably followed by irreversible transcript destabilization and decay[14–17]. Thus, the dominant effect at steady state appears to be mRNA degradation[13, 17], suggesting that analysis of transcript abundance should provide a comprehensive measure of MRE activity.

While the significance of the MRE 'seed' nucleotides in miRNA-mediated repression is undeniable[11, 18, 47, 48], demonstrating this aphorism in an endogenous cellular context, has hitherto been technically unfeasible. We show that the large repertoire of NHEJ-induced genomic deletions generated by GenERA enables direct evaluation of functional MRE sequence determinants at near-nucleotide resolution without the necessity of introducing an enrichment barcode in the targeted locus. Analysis of a subset of active MREs revealed that core seed nucleotides complementary to positions 2–7 at the 5′ end of the miRNA are indeed essential for repression of target mRNAs in their native context. In contrast, the distal nucleotides comprising the extended seed (position 1 and 8 relative to the miRNA) appeared to be dispensable for these targets or only marginally impact repression. Based on these findings, we propose that GenERA could be implemented to dissect the activity of other, less defined, intragenic cis-regulatory elements.

In principle, GenERA could enable near-nucleotide resolution analysis of any UTR, as well as de novo discovery of sequence determinants essential for RRE activity. The only intrinsic restriction for GenERA implementation is the availability and distribution of Cas9 PAM domains across a target genome. Our survey of nearly 4000 UDPs underlying the in depth analysis of CG9257 3′UTR, revealed that the median size of Cas9-induced NHEJ deletions was 19nt. To establish the broad applicability of GenERA, we calculated the distance between each nucleotide and the nearest Cas9 cut site across the entire Drosophila 3′UTR repertoire. When considering only NGG PAMs, this analysis revealed that 83% of nucleotides in Drosophila 3′UTRs are within editing range (≤19 bp) from the nearest Cas9 cut sites (Supplementary Fig. 14). Since NAG PAMs[56] were also effective in the GenERA pipeline, we carried out the same calculation including both types of PAM sites (NGG and NAG). In this instance, we found that as much as 98% of all nucleotides across Drosophila 3′ UTRs are in principle accessible for unbiased analysis by GenERA (Supplementary Fig. 14). Furthermore, SpCas9 PAM sequences are present within NHEJ editing distance of nearly all predicted miRNA target sites in various species (>95% human MREs)[30]. Finally, GenERA screens could also be carried out using new Cas9 variants with altered PAM specificities, which have vastly expanded the genome editing coverage of RNA-guided endonucleases[57–59].

Based on these considerations, we propose that GenERA brings multiplex analysis of RREs within the scope of any individual lab. Notably, the basic GenERA experimental framework is theoretically amenable to pooled lentiviral-mediated delivery of sgRNA libraries and single-cell integrated genotype/phenotype (i.e. NHEJ deletions/transcript levels) analysis by NGS[29, 60]. Such adaptations and future iterations of this technology could enable more complex high-throughput screens and potentially transcriptome-wide studies aiming to shed light on the overall contribution of post-transcriptional regulation on maintaining protein homeostasis.

## Methods

**Design and construction of GenERA sgRNAs.** sgRNAs targeting the pck control region, miR-184 MRE and polyA signal, as well as those used to build the CG9257 3′UTR mutagenesis library, were designed using CRISPR-MIT (http://crispr.mit.edu) considering both NGG and NAG protospacer adjacent motifs (PAMs). For CG9257 combinatorial MRE and miR-184 network analysis, the miR-CRISPR algorithm (http://mir-crispr.molbiol.ox.ac.uk/fulga/miR-CRISPR.cgi) was used to identify NGG and NAG PAM sequences on the plus or minus strand, and design corresponding sgRNAs in close proximity or overlapping MRE seed sequence (Supplementary Notes 1–3). Forward (fwd) and reverse (rev) sgRNA oligos were synthesized (Integrated DNA Technologies, IDT) and cloned into the pAc-sgRNA-Cas9 vector (Addgene plasmid #49330) using Bsp QI sites as previously described[33]. Briefly, 10 µl fwd and 10 µl rev sgRNA oligos (100 µM) were diluted in 20 µl 2 X Annealing buffer (20 mM Tris, 2 mM EDTA, 100 mM NaCl, pH 8.0), annealed in a thermocycler by ramping down from 95 to 25 °C at 5 °C/min, and phosphorylated in a 10 µl reaction using T4 polynucleotide kinase (NEB). The backbone plasmid (pAc-sgRNA-Cas9, 2 µg) was digested with Bsp QI (NEB), dephosphorylated (CIP, NEB) and purified using the QIAquick PCR Purification kit (Qiagen). Ligation reactions (50 ng digested pAc-sgRNA-Cas9 + 2 µl of 1:200 diluted annealed/phosphorylated oligos) were carried out at 37 °C for 1 h, and transformed into chemically competent E. coli DH5α cells. sgRNA plasmids were then isolated using the QIAprep Spin Miniprep kit (Qiagen), quantified (Nanodrop), diluted to 100 ng/µl in H₂O, and stored at −80 °C. The identity of sgRNAs was confirmed by conventional PCR using a universal U6 forward primer (5′-TCTTAAGACCATTTGCCAAT-3′) and the sgRNA rev oligos as reverse primers.

**S2R+ transfection, gDNA and RNA extraction.** S2R+ cells were cultured in Drosophila Schneider's media (Sigma) supplemented with 10% fetal bovine serum (Gibco) and 0.5% penicillin-streptomycin (Gibco) at 25 °C. Cells were plated into 6-well dishes at a density of 2 × 10⁶ cells per well. For the CG9257 3′UTR screen and the analysis of combinatorial MRE activities by multiplex GenERA, single sgRNAs or the 63 combinatorial sgRNA pools respectively (Supplementary Data 1), were individually transfected into S2R+ cells using TransIT-2020 (Mirus) at 1:5 ratio (1 µg DNA: 5 µl TransIT per well) 24 h after plating. After 48 h, puromycin (Gibco) containing media (4 µg/ml) was added to enrich for transfected cells, which were subsequently collected 10 days post-transfection All cells were washed three times with 1 ml 1xPBS at room temperature, pelleted by centrifugation (4000 g for 3 min), snap frozen and stored at −80 °C. For miR-184 network analysis, Cas9-sgRNAs (Supplementary Data 7) were individually transfected 24 h later with FuGENE HD reagent (Promega) at a 1:3 ratio (2 µg DNA: 6 µl FuGENE HD per well) according to manufacturer's recommendations. After 48 h, puromycin (ThermoFisher Scientific) containing media (4 µg/ml) was added to enrich for transfected cells. Cells were collected 96 h post-transfection, washed three times with 1 ml 1xPBS at room temperature, pelleted by centrifugation (4000 g for 3 min), snap frozen and stored at −80 °C. In all cases, genomic DNA and total RNA were simultaneously extracted from each sample using All Prep DNA/RNA Mini kit (Qiagen), quantified and diluted to the same concentration (100 ng/µl for DNA and 50 ng/µl for RNA). RNA (500 ng per sample) was reverse transcribed and genomic DNA was concomitantly removed using the QuantiTect Reverse Transcription kit (Qiagen).

**Targeted PCR.** For CG9257 3′UTR screen and multiplexed MRE analysis, a gene specific primer flanking the Illumina sequencing primer sequences (5′- GTGACTGGAGTTCAGA CGTGTGCTCTTCCGATCTCAATAACAAACAAGGTTAAT-3′) was used for cDNA synthesis to prevent any possible reverse transcription from the opposite strand. To reduce amplification bias, five individual PCR reactions were performed in parallel for each gDNA and cDNA sample as follows: Phusion high-fidelity DNA polymerase (1unit, NEB), 5X Phusion HF buffer (10 µl, NEB), gene specific fwd and rev primer (0.5 µM each), 10 mM deoxynucleotide triphosphates (dNTP) (200 µM, ThermoFisher Scientific), template DNA (250 ng), nuclease free H₂O (33.5 µl). A two-step PCR programme was used to add sequencing adapters and library indexes on each individual amplicon (Supplementary Data 2). Step I of the reaction was carried out as follows (Supplementary Data 2): 95 °C for 5 min; n x (95 °C for 20 s; custom anneal °C 30 s; 72 °C for 10 s); 72 °C for 5 min; 4 °C. Subsequently, step II was performed with amplicon-specific forward and reverse Illumina dual indexing primers (Supplementary Data 2, N = i5 and i7 Illumina indexes) as follows: 95 °C for 5 min; 10 x (95 °C for 20 s; 62 °C 30 s; 72 °C for 30 s); 72 °C for 5 min; 4 °C. To prevent any technical variations from DNA/RNA extraction, reverse transcription and PCR, four and two library replicates were independently generated for the CG9257 3′UTR screen and multiplexed MRE analysis respectively.

For miR-184 network analysis, target-specific PCR primer pairs were designed approximately 100 bp upstream and downstream of each predicted MRE seed sequence. Again, five individual PCR reactions were performed in parallel for each gDNA and cDNA sample as follows: Phusion high-fidelity DNA polymerase (1unit, NEB), 5X Phusion HF buffer (10 µl, NEB), MRE specific fwd and rev primer

(0.5 µM each), 10 mM deoxynucleotide triphosphates (dNTP) (200 µM, ThermoFisher Scientific), template DNA (250 ng), nuclease free H$_2$O (33.5 µl). Annealing temperatures and the lowest possible non-saturating cycling conditions (n ~ 20–40 cycles) were customized for each amplicon (Supplementary Data 2). Reactions were carried out in a CFX384 Touch™ thermocycler (BioRad) as follows: 95 °C for 5 min; n x (95 °C for 20 s; custom anneal °C 30 s; 72 °C for 10 s); 72 °C for 5 min; 4 °C. PCR replicates for each gDNA and cDNA sample were merged, and successful amplification within linear range was assessed by agarose gel electrophoresis.

**NGS library generation and deep sequencing**. In parallel, to establish the dynamic range and sensitivity of NGS, a calibration curve was generated as follows: 24 four-nucleotide barcoded forward primers (5′-NNNNCCAAGGGTATCGA-CAACAGA-3′) and a universal reverse primer (5′-CGATCTCGCCGCAGTAAA-3′) were used to amplify indexed RP49 amplicons from total S2R + cDNA using the following cycling conditions (95 °C for 5 min; 28 x (95 °C for 20 s; 58 °C 30 s; 72 °C for 10 s); 72 °C for 5 min; 4 °C). Samples were divided into two groups (A and B) consisting of four triplicate serial dilutions each (1x, 10x, 100x, 1000x). An equal volume (0.5 µl) of every dilution from group A was added to each replicate gDNA pool (~ 600 µl in total), and group B dilutions were similarly applied to each replicated cDNA pool.

To generate NGS libraries underlying *pck* 3′UTR analysis, *CG9257* 3′UTR screen and multiplexed MRE analysis, replicate PCR amplicons were pooled together and purified either by gel extraction or Agencourt AMPure XP beads (Beckman Coulter) (0.75X), to prevent potential primer dimer contamination. Gel purified products were subsequently cleaned up and concentrated to 10 µl total volume using the Qiagen MinElute PCR Purification Kit (Qiagen). Library QC and quantification was performed using the Qubit dsDNA HS Assay Kit (ThermoFisher Scientific), D1000 High Sensitivity Screen Tape (Agilent), and Illumina Library Quantification Kit (KAPA Biosystems). For *CG9257* analyses (3′UTR screen and multiplexed MRE analysis), 250 bp paired end sequencing was carried out on an Illumina MiSeq sequencer, to capture the entire *CG9257* 3′UTR (Illumina, MiSeq v2, 500 cycles). For the *pck* proof of concept and individual validation analyses, 150 bp paired end sequencing was performed on the same machine (Illumina, MiSeq v2, 300 cycles).

For miR-184 analysis, the relative concentration of merged gDNA and cDNA PCR products was established by agarose gel band densitometry using the ImageJ package (http://fiji.sc/Fiji). To eliminate the possibility of sequencing bias, equal amounts of each target-specific amplicon were mixed together, and two replicate library pools were independently generated for both the gDNA and cDNA amplicons. To reduce reaction volume, final gDNA and cDNA samples were passed through a QIAquick PCR purification column (Qiagen) and eluted in 50 µl buffer EB. Samples were run on 1.5% agarose gel and bands of expected size (~ 200 bp) were purified using the QIAquick Gel Extraction kit (Qiagen). To prevent biased enrichment of high GC content sequences during QIAquick extraction, gel fragments were melted in buffer PB at room temperature, instead of recommended 42 °C. A second cleanup step was then performed on a QIAquick PCR purification column to attain maximum sample purity. gDNA and cDNA deep sequencing libraries containing indexed adapters were then constructed using the NEBNext® Ultra DNA Library Prep Kit for Illumina® (NEB) and NEBNext® Multiplex Oligos for Illumina® (NEB). Library QC and quantification was performed using the Qubit dsDNA HS Assay Kit (ThermoFisher Scientific), D1000 High Sensitivity Screen Tape (Agilent), and Illumina Library Quantification Kit (KAPA Biosystems). All libraries were pooled and 150 bp paired end sequencing was performed on an Illumina MiSeq sequencer (Illumina, MiSeq v2, 300 cycles).

**Analysis of GenERA data**. The quality of pair end sequencing raw reads was first examined using the FastQC package (http://www.bioinformatics.babraham.ac.uk/projects/fastqc/). All FASTQ files were then aligned to indexed synthetic genomes using the Burrows-Wheeler Aligner[61]. The *pck* or *CG9257* 3′UTR sequences were used as reference genomes for the candidate RRE mutagenesis experiment or the unbiased 3′UTR screen and multiplexed MRE analysis, respectively. For the GenERA miR-184 network analysis, the reference genome was generated by concatenating 100 bp sequences flanking each considered miR-184 MRE (Supplementary Data 8). Subsequently, unmapped reads were removed with Samtools[62] and all mapped reads were demultiplexed into gDNA and cDNA amplicon-specific individual sam files using a custom script.

**Cumulative deletion peak maps and ROI calculations**. For every amplicon, read length signature and base alignment information strings (CIGAR strings)[62] contained within gDNA and cDNA sam files, were used to determine the percentage of reads containing deletions at each nucleotide. The resulting cumulative deletion profiles were used to derive a region of interest (ROI, defined by the beginning and end of deletion peak coordinates) using a custom-made peak-calling algorithm (Supplementary Fig. 1 Step 1). Briefly, each deletion profile curve was smoothed by applying a box blur normalized kernel, after which regions containing deletions were segmented based on changes in the signal's first derivative. The ROI was then defined as the union of the gDNA and cDNA widest region expected to contain the majority of CRISPR-mediated NHEJ deletions. Finally, automatically computed ROIs were manually verified.

**Unique deletion patterns identification and UNS calculation**. CIGAR strings associated with each sequencing read were cropped to the corresponding ROI to reduce inconsequential spurious experimental or genomic polymorphisms that could interfere with deletion-based read matching within and between gDNA/cDNA libraries (Supplementary Fig. 1 Step 2). To maximize the prevalence of high-confidence deletions, reads truncated within ROIs, CRISPR-mediated insertions, and reads containing > 10% discrete mismatches to ROI wild type sequences were discarded (Supplementary Fig. 1 Step 3). Subsequently each read was attributed a binary code representing its deletion footprint along the ROI nucleotides. For example, if AGTAG-----CAATC (where '–' = base deletion, and A = inconsequential mismatch) represents the alignment of read AGTAGCAATC to the reference sequence AGTAGTCGATCGATC, its deletion footprint code would be 000001111100000. Based on this information reads were clustered and paired between cDNA and gDNA libraries to establish an exhaustive collection of unique deletion patterns (UDPs). Subsequently, the impact of each UDP on transcript abundance was quantified by dividing its cDNA/gDNA read count ratio to the WT ratio, to generate a UDP normalized score (UNS) (Supplementary Fig. 1 Step 4).

**Classification and analysis of combinatorial editing patterns (CEPs)**. For *CG9257* combinatorial MRE analysis, FASTQ files from two replicate experiments were merged and aligned to the reference genome as detailed above. Due to the discontinuous distribution and length of deletions generated by combinatorial sgRNAs, in this instance, the ROI used to derive UDPs was defined by the full amplicon sequence for targeted NGS (362 bp, Supplementary Data 2). To increase the UNS calculation accuracy, UDPs with a gDNA and/or cDNA read count < 10 were excluded from the analysis. Each UDP was then attributed a seven character barcode reflecting the deletion status of the seven predicted MREs. For example, if an UDP covers nucleotides 95 to 130 of the ROI, the first three MREs within Zone A (k-box1, k-box2 and miR-252 spanning coordinates 92–97, 100–105, 128–133 respectively) are deleted while the last four MREs in Zone B remain intact. In this instance, the corresponding barcode, also referred to as combinatorial editing pattern (CEP), will be '- - - x x x x', where '-' stands for a deleted MRE and 'x' for an intact MRE. Using CEPs, UNS values can be clustered to derive the combinatorial effect of multiple MRE deletions on *CG9257* transcript levels. Additionally by finding complementary CEPs such that CEP1 + CEP2 = CEP3 ('- - x x x x x' + 'x x x x x - x' = '- - x x x x - x'), it is possible to assess whether combinatorial MRE deletions have an additive or cooperative effect.

**Calculation of MRE-scores for miR-184 network analysis**. To compute a consensus activity score reflecting transcript derepression imputable to MRE ablation, a weighted normalized MRE score (MRE-score) was generated by averaging UNS values derived from all high-confidence seed-deleting UDPs (deleting at least 1 nucleotide of the extended seed region (nucleotides 1–8)). Since UNS accuracy depends on the reliability of WT-scores (cDNA$_{WT}$/gDNA$_{WT}$ read count ratio), amplicons with low gDNA and/or cDNA wild-type read counts (<100 reads) were excluded from the analysis. In addition, for amplicons satisfying this condition UDPs with low gDNA and/or cDNA read counts (<10 reads) were discarded due to their tendency to generate unreliable UNS values. Similarly, an automated string search function was used to identify all UDPs generating *de novo* MREs for the most abundant S2 miRNAs (7mer seeds; Supplementary Data 3), which were excluded from the analysis due to their potential to enable artificial miRNA-mediated repression. A possible interference with assessing the repressive activity of individual miR-184 MREs could be caused by coincidental alteration of other putative RNA regulatory elements (RREs). The following putative RREs were considered based on their close proximity to miR-184 MREs (200 bp window centered on the MRE sequence): predicted target sites of most abundant S2 miRNAs (*miRanda-mirSVR* and *TargetScanFly 6.2*); AU-rich elements (*AREScore*; http://arescore.dkfz.de/arescore.pl); polyA signals (AATAAA/ATTAAA); HuR binding motifs (CISBP-RNA; http://cisbp-rna.ccbr.utoronto.ca/TFTools.php) and Pumilio response elements (TGTANA(TA)[63]). Comparative analysis of UNS distributions (Wilcoxon rank-sum test) between UDPs +/− additional putative RRE deletions revealed that most of these alterations were inconsequential to MRE-score calculation (Supplementary Fig. 10). Based on these considerations, all these UDPs were included in the MRE-score analysis pipeline. Finally, the relative contribution of each UNS (UNS$_i$) to MRE-score calculation was established based on their corresponding gDNA read count (gDNA_count$_i$):

$$\text{MRE} - \text{score} = \sum_i w_i \times \text{UNS}_i; w_i = \frac{\text{gDNA\_count}_i}{\sum_j \text{gDNA\_count}_j}$$

To establish the robustness of this computational pipeline, we compared WT-scores and UNS between all possible gDNA and cDNA library replicate pairs (Supplementary Fig. 11). This was carried out on consensus ROI coordinates computed for each amplicon from merged gDNA and cDNA files, allowing a direct comparison of UDP sets between different replicate pairs. Linear regression analysis (R *lm* function) revealed a near perfect correlation of WT-scores and UNS (Supplementary Fig. 11). Consequently, library replicates were merged to maximize UDP read counts and increase statistical power.

**Analysis of contextual features influencing MRE potency**. To explain the observed distribution of MRE-score values we assembled a set of relevant features expected to influence MRE activity. (1) Thermodynamic stability. For each target site, the minimum free energy between mature miR-184 and the corresponding MRE sequence (22 nt) was calculated using the NUPACK package[64]. (2) MRE GC content. Defines the percentage of G and C nucleotides outside the extended seed region of each MRE (positions 9–22). (3) MRE predicted secondary structure. For each MRE, the surrounding sequence (208 base pair window centered by the 8 nt extended seed) was folded using RNAplfold tool from the Vienna RNA package[65]. The accessibility of each nucleotide was then assessed based on the likelihood of itself and neighbouring 6 nucleotides being unpaired within a 100 bp folding window. To establish a baseline control for each MRE, the entire 208 bp sequence was shuffled 10,000 times to generate an average accessibility score and standard deviation at every nucleotide position. A z-score was then calculated by subtracting baseline from target sequence accessibility and dividing this value by the baseline standard deviation at each nucleotide position. Subsequently, this metric was used to assess the impact of accessibility on MRE activity. Single nucleotide z-scores corresponding to a 25 nt window upstream of the extended seed (18 nt MRE 5' + 7 nt) were found to best correlate with MRE-score groups. (4) MRE seed conservation score. 12 Flies, Mosquito, Honeybee and Beetle Multiz Alignments and phastCons Scores averaged across the 8 nt extended seed were used to analyse the sequence conservation of each MRE[66, 67]. (5) Local AU content. The AU nucleotide content was calculated over a region spanning 100 nt upstream and downstream of the extended seed regions as previously described[11]. Briefly, using a 5-nt sliding window over 200 bp centred on the seed, $N$ windows were generated. For each window $i$ the local AU content ($AU_i$) was estimated. A gobal AU score was then calculated using a weighted mean of all $AU_i$ values: each $AU_i$ was given a weight $w_i$ inversely proportional to its distance to the seed ($w_i = 1/d$ where d is the distance of the window $i$ from the seed) in order to give more weight to the windows in close proximity to the seed.

$$AU = \sum_{i=1}^{N} \frac{w_i}{\sum_j w_j} \times AU_i$$

(6) UTR contextual features. An MRE seed position index considering the distance from STOP codon to seed and the total UTR length was calculated using the following formula:

$$\text{Seed position index} = \left| \left( \frac{D}{L} \times 100 \right) - 50 \right|$$

where $D$ is the distance from STOP codon to seed and $L$ is the total length of the UTR. (7) Pol-II ChIPseq. S2 Pol-II ChIPseq raw data was obtained from Gan et al.[43]. The height of peaks proximal to the transcription start sites were quantified using the HOMER pipeline[68]. (8) Prevalence of co-targeting miRNAs. Putative target sites of most abundant S2 miRNAs other than miR-184 (miR-14, k-box miR family, bantam, miR-279 family, miR-252, miR-34, miR-282, miR-276a, miR-988, miR-317, miR-305) encoded by genes underlying the miR-184 target network were predicted using the *TargetScan 6.0* Perl script. The percentage of genes co-targeted by miR-184 and other miRNAs was then calculated across all MRE-score groups.

**miR-184 mutant generation and clonal selection**. To generate a miR-184 mutant cell line, a pair of sgRNAs (pAc-sgRNA-Cas9 vector, Addgene plasmid #49330) flanking the precursor miR-184 genomic region (*miR-184-sgRNA1* target site: 5'-CCATTGAATCGACAGGAATTCGG-3', *miR-184-sgRNA2* target site: 5'-CGAATGCACCGGCCAACCCTCGG-3') were cloned as previously described[33]. Briefly, S2R+ cells at 80% confluence were simultaneously transfected with 1 µg of each sgRNA vector using 10 µl *TransIT*-2020 reagent (1:5 ratio Mirus Bio LLC). After 36 h, puromycin supplemented media was added (5 µg/ml), and transfected cells were selected for another 48 h. Clonal selection was achieved by limiting dilution and plating of ~ 0.5 cells/well into 96 well plates containing ~ 150,000 mitomycin-treated S2R+ cells/well (incubated for 5 h with 10 µg/ml mitomycin C (Sigma) followed by washing 3x in PBS). Media was replaced twice a week for ~ 4 weeks of clonal expansion, until cells had become confluent and clonal cell growth was clearly observed. Cells were then replicated into 2 plates, one of which was used for clonal screening and the other to maintain the cell lines. Genomic DNA was extracted with fly squishing buffer: 10 mM Tris-HCl pH 8.2, 1 mM EDTA, 25 mM NaCl, and 200 µg/ml Proteinase K diluted fresh from a frozen stock (ThermoFisher Scientific). Samples were incubated at 65 °C for 30 min (protein digestion) and 95 °C for 2 min to inactivate the proteinase K. All samples were screened by conventional PCR using primers flanking the miR-184 stem-loop genomic region (miR-184F 5'-CTATTCACGCTTTAGTGCAC-3' and miR-184R 5'-CGTGGGGTAAGTATCCTCG-3') and the following cycling conditions: 95 °C for 2 min; 35 X (95 °C for 15 s; 55 °C for 30 s; 72 °C for 30 s). PCR products were surveyed by Sanger sequencing and TIDE analysis (http://tide.nki.nl, Brinkman et al.[69]). One clonal line carrying genomic deletions encompassing the miR-184 precursor sequence (*miR-184^{mut}*) was selected and validated by deep sequencing (Fig. 6c). Loss of mature miR-184 in *miR-184^{mut}* cells was confirmed by TaqMan qRT-PCR. Briefly, total RNA was extracted from wild type S2R+ and *miR-184^{mut}*

cells using the miRNeasy Mini Kit (Qiagen). RNA was reverse transcribed using the TaqMan MicroRNA Reverse Transcription Kit (Thermo Fisher Scientific). Quantitative analysis using the TaqMan MicroRNA Assay (Thermo Fisher Scientific, TaqMan microRNA probe #4427975, TaqMan Universal qPCR master mix) revealed 90% depletion of mature miR-184 miRNA (Fig. 6d).

**S2R+ cell transcriptomics profiling**. The total RNA from wild type S2R+ cells was extracted using the miRNeasy Mini kit (Qiagen) and triplicate RNAseq libraries were prepared. Briefly, RNA quantity and integrity was first assessed with Quant-IT RiboGreen RNA Assay Kit (Invitrogen) and Tapestation 2200 R6K (Agilent). 100 ng total RNA were then processed using the Ribo-Zero rRNA Removal Kit (Human/Mouse/Rat) (Epicentre/Illumina). Library preparation was performed using the TruSeq Stranded mRNA Library Prep Kit according to the manufacturer's instructions (Illumina). Libraries were size selected on Agencourt AMPure XP beads (Beckman Coulter), analyzed for size distribution on Tapestation 2200 D1K (Agilent) and quantified using SYBR® FAST Universal qPCR Kit (KAPA Biosystems). Paired end sequencing (100 bp) was performed on a HiSeq4000 platform according to company specifications (Illumina). To assess the expression levels of all active miR-184 targets (MRE-score ≥ 1; including unannotated lincRNAs), their raw read counts were extracted with Samtools and used to calculate corresponding FPKM values.

**Data availability**. All NGS datasets generated in this study are available from the NCBI sequence read archive (SRA) database under accession number PRJNA393090. The GenERA source code and computational pipeline overview have been deposited on GitHub (https://github.com) and are freely accessible under the MIT licence agreement, with the identifier DOI 10.5281/zenodo.823138.

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

## Acknowledgements

We would like to thank Jenna Schwarz and Peiyao Zhao for help with assembling the miR-184 MRE network; Ruth Williams and Maria Suciu for advice on next generation sequencing; Simon McGowan, Jelena Telenius and Nicky Gray at the Computational Biology Research Group (CBRG) University of Oxford, for assistance with processing sequencing data. Q.W. was supported by MRC (#G0902418). Q.RV.F. is supported by a Wellcome Trust PhD studentship. T.A.B. is supported by a Radcliffe Department of Medicine/MRC Scholars Programme Studentship. Y.S.M. is supported by a Clarendon Scholarship, WIMM Prize Fellowship and Christopher Welch Scholarship. D.M.V. and A.J. E. are supported by EMBL-EBI. O.H. and J.-S.K. are supported by the Institute for Basic Science (IBS-R021-D1). R.A. is supported by core funding to WTCHG (Wellcome Trust 090532/Z/09/Z) and Oxford NIHR Biomedical Research Centre. X.J. is funded by ERC Advanced Grant (340560) awarded to Ian P.M. Tomlinson. B.R.S. was supported by a MRC PhD Studentship. S.M. and G.A.H. are supported by Wellcome Trust (#105045/Z/14/Z). M. T. and T.A.M. are supported by MRC (Molecular Haematology Unit grant MC_UU_12009/ 6). P.P. was supported by a Wellcome Trust grant (#090532/Z/09/Z). N.X. was a summer student with T.A.F. A.R.B. was supported by Wellcome Trust ISSF, John Fell Research Fund (OUP), and University of Oxford Departments of Pathology, Biochemistry, Pharmacology and Physiology, Anatomy and Genetics. T.A.F. is supported by MRC (#G0902418), BBSRC (#BB/N006550/1) and Wellcome Trust ISSF (#105605/Z/14/Z).

## Author contributions

Q.W., A.R.B. and T.A.F. conceived the study and designed the experiments. Q.W. performed most of the experiments. Q.W. and T.A.B. performed the GenERA unbiased

screen and MRE cooperative studies with cloning help from N.X. A.R.B. generated the constructs for miR-184 network analysis and miR-184 knockout cell line. Q.W. and Q.R. V.F. conceived the GenERA bioinformatics pipeline with input from T.A.F. Q.R.V.F. designed and built the GenERA computational pipeline. Q.W., Q.R.V.F. and T.A.F. analyzed the results and generated the figures. R.A. and X.J. wrote the code for the cooperative study plot. D.M.V. and A.J.E. performed RNA secondary structure analysis. P.P., O.H. and J.-S.K. helped with NGS analysis. Y.S.M., T.A.B., S.M., B.R.S., M.T., G.A.H. and T.A.M. provided support and conceptual advice. Q.W., Q.R.V.F., A.R.B. and T.A.F. wrote the manuscript.

## Additional information

**Competing interests:** T.A.M. is one of the founding shareholders of Oxstem Oncology (OSO), a subsidiary company of OxStem Ltd. J.-S.K. is a co-founder and shareholder of ToolGen, Inc., a biotechnology company focused on genome editing. The remaining authors declare no competing financial interests.

