## [Peer Review File · Nature Communications]

Reviewers' Comments:

Reviewer #1:

None

Reviewer #2:

Remarks to the Author:

I have been serving as a referee for the works for the work of Qianxin Wu et al. My own comments (referee 5) resonated with those of the other referees and the work was rejected. Next, the authors decided to dissect the work in two parts and to submit two storied to Nature Communications.

The story by Qianxin Wu et al., entitled, "A CRISPR-based platform for high-content interrogation of RNA cis-regulatory elements" describes a method to characterize non coding DNA by CRISPR/Cas9.

Main concerns are about lack of sufficient novelty:

Technologies of the type reported are not novel enough. Since 2014 it is clear that CRISP-Cas9 can be utilized to screen myriad genomic permutation (e.g. Shalem et al., Science. 2014 343:84-7).

The analysis is focused on a single 3'UTR (and later a single miRNA target set) and the work falls short of demonstrating true scalability and "systems" analysis. Without scalable functional interrogation of very many 3'UTRs, the reader is left with the description of a tiling screen. Tiling screens were reported including for example in Nature Biotech 34, 192–198 2016. The same lab reported similar tools in the same journal (Nature Communications), as Bassett et al., Nat Commun. 2014 5:4640, which was an in vivo study. Therefore, progress here is incremental and is primarily in computational analysis of cDNA and gDNA NGS data.

Editorial Note: The reviewer reports mention a companion paper that was submitted to Nature Communications at the same time as this manuscript. Based on reviewer feedback, aspects of the second manuscript were incorporated into this submission.